# Charge fluctuations in the intermediate-valence ground state of SmCoIn$_5$

David W. Tam [1✉], Nicola Colonna [1,2], Neeraj Kumar [3], Cinthia Piamonteze [4], Fatima Alarab [4], Vladimir N. Strocov [4], Antonio Cervellino [4], Tom Fennell [1], Dariusz Jakub Gawryluk [5], Ekaterina Pomjakushina [5], Y. Soh[3] & Michel Kenzelmann [1✉]

The microscopic mechanism of heavy band formation, relevant for unconventional super-conductivity in CeCoIn$_5$ and other Ce-based heavy fermion materials, depends strongly on the efficiency with which $f$ electrons are delocalized from the rare earth sites and participate in a Kondo lattice. Replacing Ce$^{3+}$ ($4f^1$, $J = 5/2$) with Sm$^{3+}$ ($4f^5$, $J = 5/2$), we show that a combination of the crystal electric field and on-site Coulomb repulsion causes SmCoIn$_5$ to exhibit a $\Gamma_7$ ground state similar to CeCoIn$_5$ with multiple $f$ electrons. We show that with this single-ion ground state, SmCoIn$_5$ exhibits a temperature-induced valence crossover consistent with a Kondo scenario, leading to increased delocalization of $f$ holes below a temperature scale set by the crystal field, $T_v \approx 60$ K. Our result provides evidence that in the case of many $f$ electrons, the crystal field remains the dominant tuning knob in controlling the efficiency of delocalization near a heavy fermion quantum critical point, and additionally clarifies that charge fluctuations play a general role in the ground state of "115" materials.

[1] Laboratory for Neutron Scattering and Imaging, Paul Scherrer Institut, 5232 Villigen, Switzerland. [2] National Centre for Computational Design and Discovery of Novel Materials (MARVEL), Ecole Polytechnique Fédérale de Lausanne, 1015 Lausanne, Switzerland. [3] Paul Scherrer Institut, 5232 Villigen, Switzerland. [4] Photon Science Division, Paul Scherrer Institut, 5232 Villigen, Switzerland. [5] Laboratory for Multiscale Materials Experiments, Paul Scherrer Institute, 5232 Villigen, Switzerland. ✉email: david-william.tam@psi.ch; michel.kenzelmann@psi.ch

Understanding the ground states that result from hybridization between local moments and conduction electrons remains a major goal of the condensed matter community. In a purely magnetic picture, localized $f$ magnetic moments can hybridize with the itinerant metallic $spd$ electrons in the valence band, forming spin singlets and resulting in a screening of the $f$ magnetic moments that reduces the tendency toward magnetic order. Within this picture, introducing a Kondo coupling $\mathcal{J}$ between the local $f$ moment and the conduction electron $c$, one traverses the well-known Doniach phase diagram between an RKKY-mediated magnetically ordered ground state ($\sim \mathcal{J}^2$) and a nonmagnetic heavy fermi liquid ($\sim e^{-1/\mathcal{J}}$), separated by a quantum critical point (QCP). Microscopically, the hybridization proceeds as the formation of a virtual bound state (Abrikosov-Suhl resonance) between the mobile $c$ carriers at the Fermi energy $E_F$, and the local $f$ electrons, with the bare interaction potential $V_{cf}$ enhanced based on how far the $f$ levels happen to be from $E_F$. For nearby $f$ states, the resonant enhancement $\mathcal{J} = V_{cf}/(E_F - E_{4f})$ leads to an increased density of heavy fermion states at $E_F$[1–3]. It is the strength of the Kondo coupling which determines whether the $f$ electrons can form a nonmagnetic Fermi liquid with heavy bands containing the $f$ electrons, which can host unconventional superconductivity. Experimental evidence for a lattice of heavy Kondo bands is observed in Ce-based materials such as CeCoIn$_5$[4–6], CeIrIn$_5$[7], and CeRh$_2$Si$_2$[8].

The virtual bound state of the Kondo lattice is generally considered to be a singlet state of the magnetic spins, but $cf$-hybridization between conducting carriers $c$ and local $f$ moments in general also allows for fluctuating charge degrees of freedom. When $f$ electrons are strongly localized in a metallic host, the charges may be conceptualized as a kind of Wigner crystal with high electronic repulsion between neighboring $f$ states, which prevents them from moving, similar to the physics of a Mott insulator[2,9]. When the primary valence configuration becomes unstable, however, charge fluctuations and ultimately delocalization of the $f$ electrons may also occur, which result in a superposition of $f^n$ and $f^{n\pm1}$ valence configurations of the localized ion[9]. If both $f$ valence states lie within the bandwidth of the metallic $spd$ electrons, one of the $f$ configurations may then be found close enough to the Fermi energy to form an Abrikosov-Suhl resonance, an effect that has been studied with a high degree of detail in systems such as YbRh$_2$Si$_2$[10]. In intermediate valence systems, the Kondo model is not necessarily able to capture the essential physics due to the existence of the charge fluctuation degree of freedom.

Distinguishing whether charge fluctuations play a role in the mechanism of unconventional superconductivity is generally difficult, but the shape of the superconducting dome has historically provided some insight. Specifically, small superconducting domes may be magnetically driven, whereas large superconducting domes extending far from the QCP and exhibiting unusual shapes are assumed to include charge fluctuations[11]. In CeCoIn$_5$, a tetragonal material, the microscopic mechanism for superconductivity was often believed to be spin-dominated due to its much larger critical temperature $T_c$ compared to CeIn$_3$, a scenario consistent with the expectations of spin-fluctuation driven superconductivity in a system with reduced dimensionality[12]. Nevertheless, a very recent study argued that CeCoIn$_5$ also exhibits a simultaneous charge delocalization QCP, which was suggested as evidence for the separation of spin and charge sectors on a more equitable footing[13], a conclusion that agrees with an earlier review of critical valence fluctuations[14]. Thus, the microscopic details of the delocalization mechanism from the point of view of the charge sector may deserve more attention in "115" materials. From this perspective, materials with multiple $f$ electrons are highly suitable for such an investigation because both valence states will have a nonzero number of $f$ electrons.

The microscopic mechanism of $f$ electron delocalization is a topic of ongoing debate, and which has invited a variety of perspectives on its origin and appropriate theoretical treatment. In the case of a single $f$ electron or hole, such as in Ce and Yb, materials with an intermediate-valence scenario are often viewed as the natural consequence of a "Kondo interaction with high Kondo temperature," which progressively brings the $f$ state closer to the delocalized limit as the temperature is lowered. However, in the multi-$f$-electron case and sometimes including Yb, it has been recognized that this picture misses some additional complexity[2,15]. Starting with basic experimental observations, the $f$ delocalization mechanism seems to be enhanced in ions which already have a tendency toward an $f$ valence instability, including Ce, Sm, Yb, U, and Pu[3]. In 5$f$ intermediate-valence materials with U and Pu, theoretical efforts have been made to identify an integer number of delocalized $f$ electrons based on stability conditions[16–18], while for 4$f$ materials with Ce and Yb, as well as Sm-based materials like SmB$_6$, the valence is often well-established to be a fraction of a single $f$ electron[19,20]. In YbCoIn$_5$, it was argued that the intermediate-valence state does not change with temperature, meaning Yb acts as a magnetic impurity similar to the role of Nd in Ce$_{1-x}$Nd$_x$CoIn$_5$[21] and suggesting a non-Kondo origin for the intermediate-valence character[19]. A particularly intriguing mystery was found by comparing different Yb-based materials, where delocalization through a intermediate valence mechanism occurs at a fixed temperature scale, even when the Kondo temperatures differ by over 4 orders of magnitude[22]. For Sm-based compounds, modern experiments have shown that they often exhibit heavy fermion and/or intermediate valence character[9,23–34]. Of these cases, a temperature-induced Kondo effect was most clearly observed in (La,Sm)Sn$_3$, with a Kondo temperature near 125 K in the dilute case, and a Kondo lattice in the dense (Sm-rich) limit at a similar temperature scale[25,35,36]. In SmOs$_4$Sb$_{12}$, on the other hand, an exotic low-temperature heavy fermion state was suggested to be non-magnetically driven, and valence fluctuations are also observed, suggesting a novel type of heavy fermion mass enhancement for the multi-$f$-electron case[27,34]. A key observation about the origin of heavy masses, pointed out by Higashinaka et al.[37], is that Sm-based systems do not follow the expected mass enhancement factor $m^*/m = (1 + \Delta n_f)/(2\Delta n_f)$. This factor, calculated to be the dominant scale factor in the renormalized quasiparticle mass, reflects the decrease in effective mass when the itinerancy of the $f$ states is large, and has been empirically shown to work well for systems with a single $f$ electron in the strongly correlated limit[14]. However, Sm-based systems tend to show a large mass enhancement of $10^2–10^3$ when they exhibit a valence of $v_{Sm} \approx 2.8$ ($\Delta n_f = 0.2$), whereas the formula predicts only $m^*/m = 3$. Therefore, since the phenomenology of mass enhancement in Sm-based materials is very different from expectations, it is important to show how $f$ electrons in Sm-based materials can exhibit temperature-dependent delocalization in order to understand how Kondo-like behavior arises within the many-body intermediate valence scenario.

Turning now to SmCoIn$_5$, we present evidence for an $f$ electron delocalization mechanism at temperatures below $T_v \approx 60$ K, which we identify as a valence crossover temperature. In Fig. 1, we draw a cartoon schematic that showcases our understanding of the valence change that is coincident with a change in the shape of the Sm 4$f$ wavefunctions. Figure 1a shows one layer of Sm$^{3+}$ ions in SmCoIn$_5$ at high temperature, where all of the levels in the ground spin-orbit multiplet are thermally populated, and the total

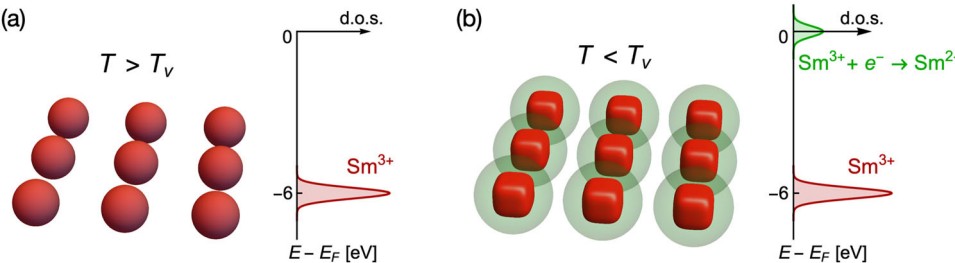

**Fig. 1 Cartoon schematic of the *f* delocalization process in SmCoIn$_5$.** **a** Lattice of Sm$^{3+}$ ions at high temperature above the valence transition $T_v \approx 60$ K, with a graph showing the position of Sm$^{3+}$ in the density of states (d.o.s.) with respect to the Fermi energy $E_F$. **b** Lattice of Sm$^{3+}$ ions at low temperature below $T_v$, but above the antiferromagnetically ordered phase at $T_N \approx 11$ K, containing screening clouds around each site representing the delocalized Sm$^{2+}$ states that appear in the d.o.s. near $E_F$.

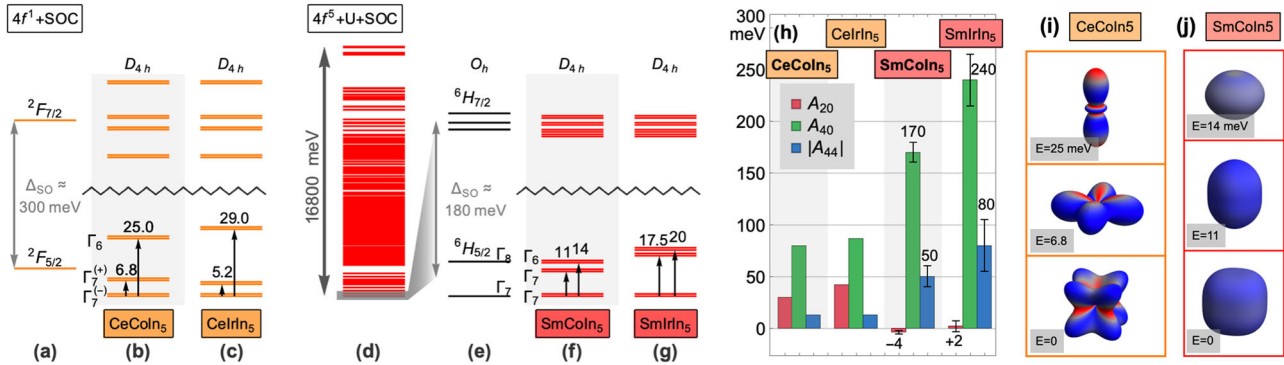

**Fig. 2 Single-ion properties of Ce$^{3+}$ and Sm$^{3+}$ in the "115" materials. a–g** Level scheme of the Ce$^{3+}$ and Sm$^{3+}$ ions in CeCoIn$_5$ and SmCoIn$_5$, calculated using the software package QUANTY. The ground manifold in each case consists of three Kramers doublets with $J \approx 5/2$, separated from the 7/2 manifold by spin-orbit coupling as indicated by the zigzag lines. The CeCoIn$_5$ calculations in **a–c** were performed for the $4f^1$ configuration using the crystal field parameters reported by Willers et al.[39] and spin-orbit interaction $\xi_{SO} = 87$ meV, which leads to a splitting between the $J = 5/2$ and $J = 7/2$ spin-orbit manifolds of $\Delta \approx 300$ meV. The SmCoIn$_5$ calculations in **d–g** were performed using the parameters found in this work and with $\xi_{SO} \approx 102$ meV, which leads to a lower $\Delta \approx 180$ meV. **h** Stevens A parameters for the crystal field potential used in the calculations, in meV: $(-4, 170, 50)$ for SmCoIn$_5$ and $(2, 240, 80)$ for SmIrIn$_5$, which were found via an exhaustive search in the space of $A_2^0$, $A_4^0$, and $A_4^4$. For SmCoIn$_5$, the error bars are chosen by hand to account for reasonable variation in the parameters that still remains a close match to the experimental data (XLD and magnetic susceptibility) and reproduce the temperature ($T_v = 60$ K) at which the XLD pattern inverts. For SmIrIn$_5$, the error bars were chosen to match INS and magnetic susceptibility. **i–j** Ground state wavefunctions (one from each Kramers doublet) of CeCoIn$_5$ and SmCoIn$_5$, plotted with QUANTY. The radius corresponds to the magnitude of the charge density, with magnetization density overlaid as the surface color from red to blue. For CeCoIn$_5$ and CeIrIn$_5$, the values were converted from the Stevens B parameters reported by Willers et al.[39] using the conversion factors $(-35, 1260, 18\sqrt{70})$[72].

$4f$ wavefunction at each site resembles a sphere. At temperatures below $T_v$, the Sm$^{3+}$ wavefunctions become thermally isolated in the ground state doublet, which have a box-like shape, while delocalized states corresponding to the Sm$^{2+}$ valence configuration are enhanced at the Fermi energy $E_F$, as depicted in Fig. 1b. Evidence for this scenario comes primarily from our x-ray absorption spectroscopy (XAS) experiments, combined with full multiplet calculations, which reveal the microscopic configuration of $f$ electrons in the Sm$^{3+}$ valence state of SmCoIn$_5$ and determine the energy level scheme (Figs. 2 and 3). Further evidence for a valence crossover $T_v$ associated with this crystal field scheme is observed with a variety of experimental probes (Fig. 4), including the onset of magnetic fluctuations in bulk susceptibility, the appearance of magnetism in the Co $d$ electron orbitals, a crossover in the lattice parameter aspect ratio $a/c$, a departure from $T$-linear resistivity and inflection point in carrier density, and most significantly, a redistribution of $f$ spectral weight in the valence band structure favoring a Sm$^{2+}$ component around the Fermi energy $E_F$ at low temperature, which represents a delocalized hole in the conduction volume (Fig. 5). These results indicate that $T_v$ is related to a change in the microscopic shape of the Sm wavefunctions that allows electrons in the Fermi sea to spend more time on the Sm sites, resulting in a larger Sm$^{2+}$ signature. In addition, our full multiplet calculations (Figs. 2 and 3) show that

the mutual Coulomb repulsion between $f$ electrons is responsible for separating the Sm single-ion energy eigenstates over more than 16 eV in the valence band. Such a wide distribution of multiplet levels, containing wavefunctions with many different shapes and symmetries, may point to the importance of one of these levels serving as a virtual intermediate state in the formation of an Abrikosov-Suhl resonance and heavy fermion character. The appearance of $T_v \approx 60$ K in bulk probes (magnetization, lattice constant, resistivity, and carrier density) also shows that the crossover is a bulk property, with possible consequences for the eventual magnetically ordered ground state below $T_N \approx 11$ K[38]. Therefore, our work shows that SmCoIn$_5$ exhibits a delocalization mechanism driven by the crystal field that resembles the Kondo effect in CeCoIn$_5$ generalized to the multi-$f$-electron case, and shows that charge fluctuations are an important phenomenon found widely in the ground state of "115" materials.

## Results

### Crystal field ground state and multiplet structure of SmCoIn$_5$ and SmIrIn$_5$.

The Hamiltonian of the localized $4f$ electrons may be written as $H = H_{Coul} + H_{SO} + H_{CEF}$, which contains the Coulomb interaction, spin-orbit coupling, and crystal electric field with $D_{4h}$ site symmetry. In CeCoIn$_5$, the crystal field was

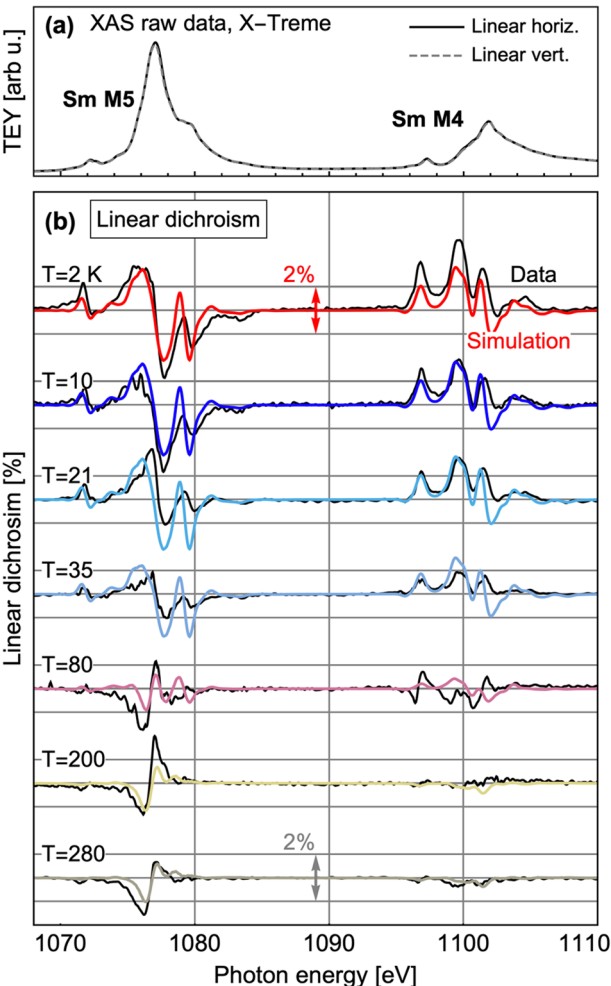

**Fig. 3 XAS spectra and model calculations for SmCoIn₅. a** XAS at the Sm $M_{4,5}$ edges, measured at X-Treme in grazing incidence using the total electron yield (TEY) method. **b** XLD as a function of temperature, and corresponding QUANTY calculations for the Sm³⁺ ions. The XLD data is normalized to the XAS signal, with a vertical scale of 2% shown with horizontal lines around each curve, as indicated by double-headed arrows. The overall agreement between the data and model calculations at all temperatures is sufficient to uniquely determine the Stevens parameters presented in Fig. 2.

fully determined from measurements of the x-ray linear dichroism (XLD) in x-ray absorption spectroscopy (XAS) experiments at the Ce $M_{4,5}$ edges, combined with inelastic neutron scattering (INS)[39–41]. We determined the single-ion properties of Sm in SmCoIn₅ by combining $M_{4,5}$-edge XLD with bulk magnetization measurements. In order to model the XLD results, we used the full-multiplet software package QUANTY[42] to diagonalize the full Hamiltonian for the $4f$ ground state. With $D_{4h}$ site symmetry and 5 electrons in the $f$ shell, there are three independent parameters in the crystal field potential expansion written in Stevens operator formalism, specifically, a dipole term $A_2^0$ which describes the ionic potential along the long tetragonal axis, and two quadrupolar terms $A_4^0$ and $A_4^4$.

The results of our analysis are summarized in Fig. 2, which shows both Sm compounds exhibit a $\Gamma_7$ ground state wavefunction, similar to the results previously found for the Ce-based 115s. We find the crystal field scheme for Sm is very close to the limit of cubic $O_h$ symmetry, where the tetragonal terms are constrained to a fixed ratio and $A_2^0 = 0$[43,44]. Upon the lowering of the point

group symmetry from cubic to tetragonal, the $\Gamma_7$ doublet survives while the $\Gamma_8$ cubic quartet separates into $\Gamma_6$ and $\Gamma_7$ doublets[45]. In SmCoIn₅, we find a very small dipolar $A_2^0$ term, as well as a ratio close to the cubic limit for the quadrupolar terms, which implies that the tetragonal anisotropy of the lattice has a weak effect on the single-ion properties of Sm. Within this crystal field scheme, the Sm³⁺ level spacing and wavefunctions are significantly affected by the Coulomb repulsion between mutual $f$ electrons (see Supplementary Note 1 and Supplementary Fig. 1), a situation not encountered in the single-electron case. Moreover, we find that the Coulomb repulsion admixes the ground state wavefunction with other multiplets, resulting in a situation where the combination of irreducible representations contained in the ground state wavefunction is not perfectly described by mixing the $J_z = 5/2$ and $3/2$ states, as it is in the single-electron Ce³⁺[46]. In Table 1, we compute the expectation values of quantum angular momentum operators of the ground state, as well as the projection into the irreducible symmetries of the $D_{4h}$ point symmetry group, under different combinations of the terms in the Hamiltonian. Introducing spin-orbit coupling (SOC), the crystal field potential, and Coulomb repulsion have different effects on the quantum numbers, indicating that the ground state of Sm is not in the limit of a pure $LS$ multiplet state.

In Fig. 3, we present the x-ray linear dichroism (XLD) raw data, displayed as the $c$-axis component of the XAS measurement minus the $a$-axis component (see Methods). By examining the XLD data above and below a temperature near $T_\nu \approx 60$ K, the shape of the dichroism reverses sign. Since the XAS spectra include transition processes that span several eV, and each transition depends on the Boltzmann factor of its initial state, the most obvious qualitative crossover of the overall XLD lineshape will occur when the system transitions between an ensemble of populated crystal field states to a pure doublet state at low temperature, which is precisely the crossover temperature $T_\nu \approx 60$ K that we find. Specifically, we found that the dipole $A_2^0$ parameter is coupled strongly to the temperature of this sign reversal; this allows us to calculate $A_2^0$ in a way that is highly robust against small errors in the raw data, giving us confidence in our analysis. The model calculations in Fig. 3 are generated with QUANTY and are calculated for the Sm³⁺ configuration. For the Sm²⁺ configuration, the $4f^6$ configuration exhibits $J = 0$ with no low-lying crystal field states, and is a non-Kramers ion; therefore, we ignore the Sm²⁺ states in the analysis of the XAS data and magnetization.

Since our crystals of SmCoIn₅ are small, only around 10 mg, we were unable to perform inelastic neutron scattering (INS) to confirm the crystal field level spacing. Thus, in order to roughly confirm the results for SmCoIn₅, we also synthesized SmIrIn₅, obtaining large crystals which we prepared in a powder sample for INS with a total mass of 1.8 g (see Methods). We conducted constant-Q scans at six positions between Q = 1.79 and 4.00 Å⁻¹, which all show a broad peak that is consistent with two overlapping resolution-limited peaks near E = 17.5 and E = 20 meV, suggesting that the two crystal field doublets from Sm are at these positions (see Supplementary Note 3 and Supplementary Fig. 3). This level scheme for SmIrIn₅ is also consistent with the magnetization data we collected on a 63 mg single crystal of SmIrIn₅ at high temperature (see Supplementary Note 4 and Supplementary Fig. 4), which shows that the crystal field in SmIrIn₅ is close to the cubic limit such that $A_2^0 \approx 0$, as we found with SmCoIn₅. Therefore, by combining magnetization data with with inelastic neutron scattering for the SmIrIn₅ compound, as a complementary technique to combining magnetization with XAS experiments as we did for SmCoIn₅, and finding a similar level scheme in the ground multiplet for both Sm-based materials,

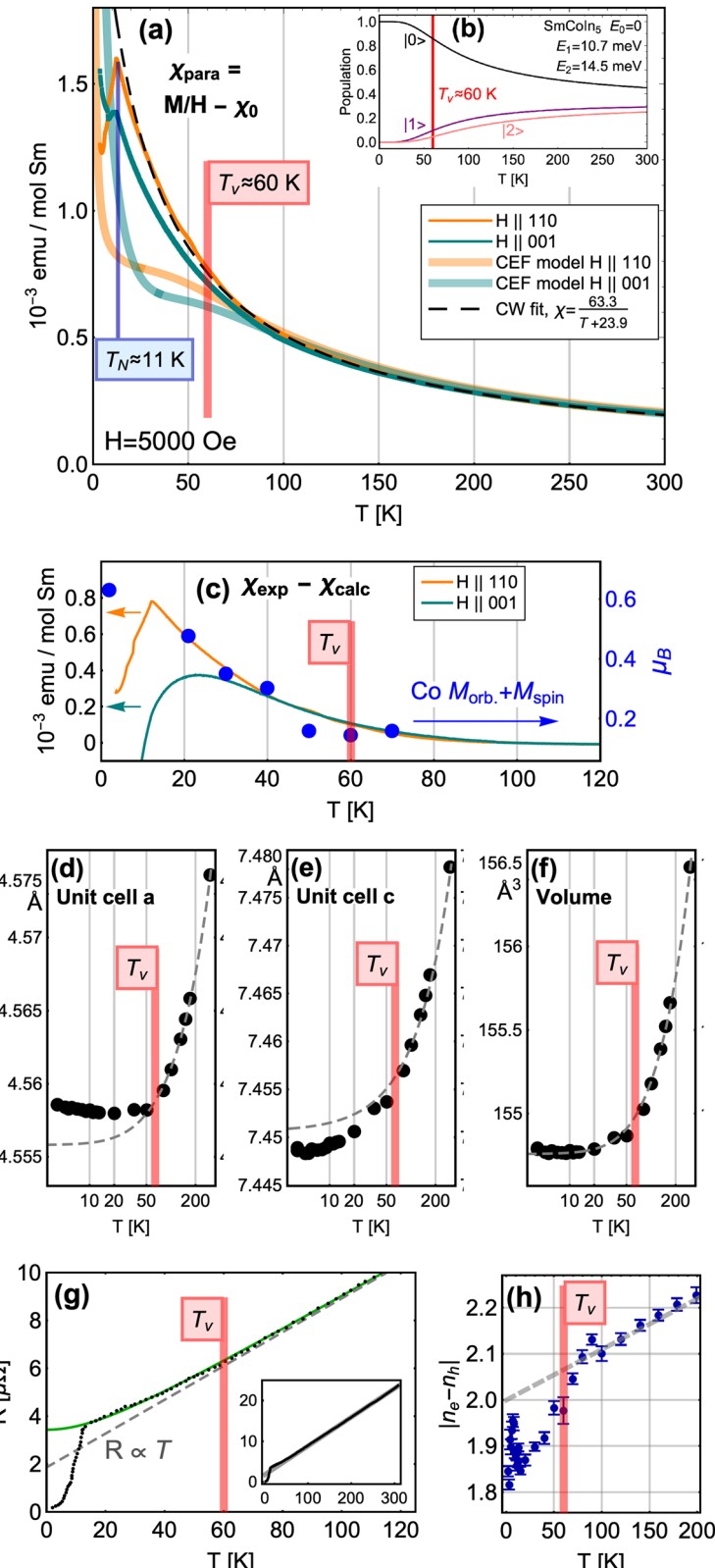

gives us confidence that the crystal field ground state of SmCoIn$_5$ and SmIrIn$_5$ are accurately determined.

**Temperature scale near $T_v \approx 60$ K.** In Fig. 4, we show the evidence for a crossover in the electronic character of SmCoIn$_5$ near $T_v \approx 60$ K. Magnetization measurements of SmCoIn$_5$ are shown in

Fig. 4a, along with the results of the calculated magnetic susceptibility within our crystal field model of Sm$^{3+}$, for magnetic field of 5000 Oe applied in different directions, within the $ab$ plane and parallel to the tetragonal $c$ axis. The magnetization contains no contribution from the $J = 0$ Sm$^{2+}$ states. Near $T_v$, we find that the calculated magnetization exhibits a hump as a function of temperature, and below $T_v$ begins to deviate from the

**Fig. 4 Evidence for a temperature scale $T_v \approx 60$ K in SmCoIn$_5$. a** Magnetic susceptibility of SmCoIn$_5$ measured in a vibrating sample magnetometer with H = 5000 Oe, shown along with calculations of the magnetization using the crystal field model in QUANTY, and with a Curie–Weiss fit. The magnetic ordering temperature is visible at $T_N = 11$ K. **b** (inset) State populations of the three multiplet levels of SmCoIn$_5$, showing an inflection point in the ground state population at T = 56 K. The equivalent temperature for CeCoIn$_5$ is found to be T = 36 K. **c** Difference between the experimentally observed magnetization and calculated magnetization for different field directions. The overlayed points are the total magnetic moment of the conduction electrons at H = 6.8 T with H||101, which we obtained from sum rule analysis of x-ray magnetic circular dichroism (XMCD) data measured at the Co L-edge assuming a Co$^{2+}$ valence (see Supplementary Note 4 and Supplementary Figs. 5–7). **d–f** Temperature dependence of the $a$ and $c$ lattice parameters, and corresponding fits to the high temperature data with a hyperbolic form $a_i = a_{i,0} + b\sqrt{(T/T^*)^2 + 1}$, where $b$ and $T^*$ are fitted parameters with error bars smaller than the point size in the figure. The departure from the expected form occurs near $T_v$. **g** Direct current electrical resistance showing a departure from linear behavior at $T_v$. The solid line shows a fit to a hyperbolic form. Inset: resistance continues in a linear fashion up to T = 300 K. **h** Net carrier density from Hall resistance measurements with current applied along the $a$ axis direction and field along $c$. Error bars represent fitted errors.

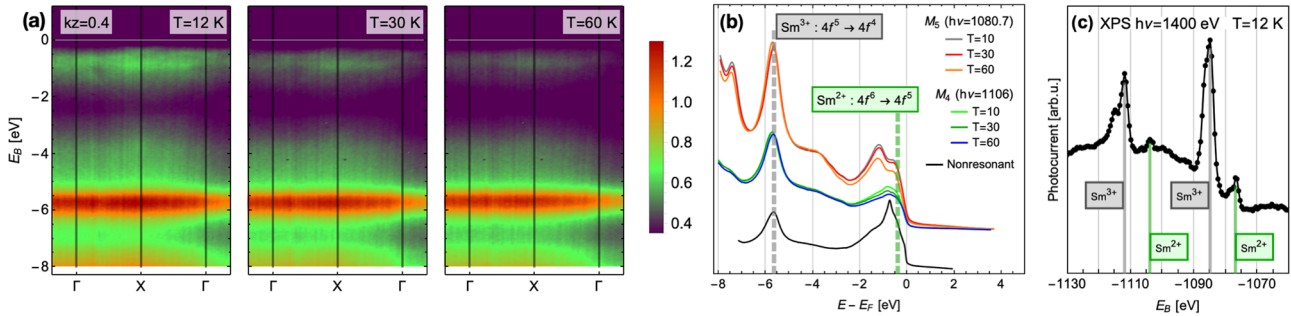

**Fig. 5 Evidence for intermediate valency in SmCoIn$_5$ below $T_v \approx 60$ K. a** Temperature dependence of the momentum-resolved Res-ARPES spectra along the $\Gamma - X - \Gamma$ high-symmetry cut at the Sm $M_4$ edge ($h\nu = 1106$ eV), showing an enhancement of the spectral weight near $-2 < E_B < -0.5$ eV that is associated with the appearance of the Sm$^{2+}$ configuration (see Supplementary Note 2 and Supplementary Fig. 2). The emission patterns are consistent with the multiplet spectrum of Sm final states in the resonant ARPES process[52,53]. **b** Momentum-integrated cuts along $\Gamma - X - \Gamma$ as a function of binding energy, at the Sm $M_{4,5}$ resonant edges, for several temperatures below $T_v \approx 60$ K. The peak structure near $-2 < E_B < -0.5$ eV associated with Sm$^{2+}$ loses spectral weight as the temperature is increased. A nonresonant ARPES measurement carried out at T = 12 K with $h\nu = 605$ eV and integrated over the same momentum cut is shown offset at the bottom, which also contains Co and In bands. **c** X-ray photoemission spectroscopy at T = 12 K, showing an intermediate valence scenario.

experimental results. To better understand the origin of the temperature $T_v = 60$ K, in the inset labeled Fig. 4b we plot the Boltzmann population factors for the levels found from the XLD data shown in Fig. 3. The inflection point in the population factor of the ground state is calculated to be 56 K, which is an excellent match for the observed $T_v$ in all experimental probes and strongly suggests that the crystal field is the driving mechanism behind the observation of $T_v$. Moreover, the experimental magnetization data show the $f$ magnetic moments in SmCoIn$_5$ do not follow the expected magnetic susceptibility from the crystal field model, but instead follow almost perfect Curie–Weiss behavior with Weiss temperature $\Theta \approx -24$ K and $J \approx 2.04$. Below $T_N = 11$ K, the susceptibility exhibits a sharp change due to the onset of long-range antiferromagnetic order. The departure of the magnetic susceptibility from the crystal field model of Sm$^{3+}$ suggests that the magnetic degree of freedom is not strongly affected by the valence crossover and change of shape of the 4$f$ atomic wavefunction.

To better understand the meaning of the departure from the crystal field model below $T_v$, we measured x-ray magnetic circular dichroism (XMCD) at the Co L-edge to understand whether the magnetic susceptibility arises from the conduction electrons. In previous experiments, it was determined that the magnetic moment specifically tied to the conduction electrons could be measured with XMCD at the ligand ions of UAs, UGa$_3$, and UGe$_2$[47,48]. Here, we perform the same analysis, while also considering that the Co 3$d$ states may have a small intrinsic moment. Using the XMCD sum rules[49], we determine that below $T_v$, the conduction electron moment is increasing in proportion with the departure of the bulk magnetic susceptibility from the crystal field model, gaining approximately 0.3 $\mu_B$ between $T_v$ and

$T_N = 11$ K. In Fig. 4c, we overlay the total magnetic moment of Co onto the difference curve between bulk magnetization and our crystal field model from part (a). The agreement between the bulk and local probe experiments demonstrates that itinerant electrons carry part of the $f$ electron magnetic moment below $T_v$, a signature of delocalization.

To further demonstrate a change of physical properties across $T_v$, we used high-resolution powder x-ray diffraction experiments to search for changes in the structure of the lattice. In Fig. 4d–f, we show that the $a$ and $c$ tetragonal lattice parameters both change qualitatively below $T_v$ with respect to the temperature dependence of the total unit cell volume $a^2/c$. The total volume follows the form of a hyperbola, which smoothly interpolates between constant and linear thermal expansion as the temperature is increased. However, the individual lattice parameters both deviate from a hyperbolic form below $T_v$, with the $a$ lattice parameter even beginning to increase as the temperature is lowered further, indicating an unexpected in-plane expansion of the unit cell. This result shows phenomenologically that the electronic structure undergoes a crossover transition at $T_v$, and proves that the transition occurs in the bulk of SmCoIn$_5$. Moreover, the direct cause of the change in the aspect ratio $a/c$ may be connected to magnetostrictive effects, due to the partial transfer of the magnetic moments onto the itinerant sites (Fig. 4a, b). In this way, the lattice parameter measurements show that the valence crossover also affects the bulk magnetic properties and leads to structural changes, while the overall tetragonal symmetry is preserved. It is also possible that $T_v$ is associated with the development of an intrinsic Co moment, which may also be related to the overall change in electronic structure connected to the valence transition at $T_v$.

**Table 1 Angular momentum quantum numbers of the crystal field ground states (g.s.) of Sm$^{3+}$ in SmCoIn$_5$ calculated using QUANTY.**

| Config. | Energy | $L^2$ | $S^2$ | $J^2$ | $L_z$ | $S_z$ | $J_z$ | $N_{A2u}$ | $N_{B1u}$ | $N_{B2u}$ | $N_{Eu1}$ | $N_{Eu2}$ |
|---|---|---|---|---|---|---|---|---|---|---|---|---|
| SOC only | 0 (g.s.) | 14.2857 | 3.03571 | 8.75 | −2.85714 | 0.357143 | −2.5 | 0.857143 | 0.785714 | 0.785714 | 1.17857 | 1.39286 |
| SOC + CEF | 0 (g.s.) | 14.1873 | 3.01221 | 8.82941 | −0.590664 | 0.091235 | −0.499429 | 0.408765 | 0.921217 | 0.888367 | 1.44416 | 1.33749 |
| SOC + CEF + U | 0 (g.s.) | 29.3049 | 8.40893 | 8.79379 | −2.21299 | 1.00527 | −1.20771 | 0.633231 | 0.845672 | 0.76524 | 1.32723 | 1.42863 |

The parameters used are: $\xi_{SOC} = 180$ meV (spin-orbit coupling), Stevens $A$ parameters $A_2^0 = −4$ meV, $A_4^0 = 170$ meV, and $A_4^4 = 50$ meV for the crystal electric field (CEF), and Coulomb coupling ($U$) at 80% of the Cowan code values, with a magnetic field of 10 Gauss used to obtain the quantum numbers numerically. The additional columns show the expectation values of the number operator $N$ for the five symmetry-adapted irreducible representations of the $D_{4h}$ point group.

Finally, in Fig. 4g, h we show a qualitative change in the electrical resistance and carrier density which occur below $T_\nu$. In the temperature-dependent resistance R(T) at temperatures above $T_\nu$ (inset of Fig. 4g), the resistivity is clearly linear in T, a phenomenon previously associated with the existence of critical valence fluctuations[14]. Below $T_\nu$, the resistance begins an "upturn," which keeps the value higher than the T-linear model predicts as the temperature is lowered. This observation is consistent with opening more scattering channels for the conduction electrons and holes due to the increase in the number of Sm$^{2+}$ states at $E_F$[50]. A similar "upturn" in resistivity measurements of SmSn$_3$ was taken as a sign of the Kondo effect[25].

To further show that the upturn in resistivity is connected to a qualitative change in the metallic properties of SmCoIn$_5$, we carried out measurements of the Hall effect as a function of temperature, and extracted the net total carrier density from a model fitted to data between $H = −9$ and $+9$ T. With current along the a-axis direction and magnetic field of H = 9 T applied along the c-axis, we measured the carrier density as a function of temperature between T = 2 and 200 K. The result shows that the carrier density also exhibits a qualitative decrease below $T_\nu \approx 60$ K, consistent with the introduction of more hole-like states in the carrier volume. This simultaneous increase in hole carriers and qualitative change in R(T) shows that the resistivity "upturn" in Fig. 4g is most likely associated with an increased number of scattering channels below $T_\nu$ due to the presence of more Sm$^{2+}$ states. Therefore, bulk resistivity measurements provide evidence for a valence crossover in SmCoIn$_5$ at $T_\nu$.

**Valence crossover observed with resonant ARPES**. To further show that $T_\nu \approx 60$ K is associated with hybridization of the Sm $f$ electrons with conduction states at the Fermi level, we measured the position and intensity of the Sm$^{3+}$ and Sm$^{2+}$ configurations in the valence band using resonant angle-resolved photoemission (Res-ARPES) at the Sm $M_{4,5}$ edges. In the resonant process, photons may excite electrons from the Sm $d$-shell into the vacuum through an intermediate state in the valence band. Since the intermediate state is short lived, its energy width is large by the uncertainty principle, and therefore any available Sm $f$ state around the valence band may be temporarily occupied. This means that Res-ARPES observes resonant emission at the positions of any multiplet satisfying the appropriate photon selection rules, regardless of whether it is thermodynamically occupied, generating a "fingerprint" of different multiplet bands that can be used to identify the binding energy of different valence states. Analysis using the Res-ARPES spectrum has been carried out previously in several studies of multi-$f$-electron systems, including YbCuIn$_4$[51] and SmB$_6$[23,51,52].

The Res-ARPES raw data at different temperatures is shown in Fig. 5a, with momentum-integrated cuts shown in Fig. 5b as a function of binding energy $E_B$. The valence states of Sm are Sm$^{3+}$ ($4f^5$) and Sm$^{2+}$ ($4f^6$), which after emitting a photoelectron are in the $4f^4$ and $4f^5$ final state configurations, respectively, which are the states observed in the experiment. The multiplet spectrum from each valence shows the position of the initial state, because

the ground multiplet of the final state is energetically degenerate with the initial state. In our DFT calculations of SmCoIn$_5$, we find excellent agreement for the position of the Sm$^{3+}$ states at $E_B \approx 6$ eV (see Supplementary Note 2 and Supplementary Fig. 2), with the Sm$^{2+}$ states appearing close to the Fermi energy, which is in agreement with the Res-ARPES data. The intensity and "fingerprint" of multiplet emission lines for each valence state were tabulated by Gerken[53] and also agree with our observations. Moreover, the resonant spectrum in SmCoIn$_5$ at low temperature appears to be highly similar to that of intermediate-valence SmB$_6$[52]. In SmCoIn$_5$, we find that the Sm$^{3+}$ band maintains similar intensity as a function of temperature between T = 12 and 60 K, whereas the Sm$^{2+}$ component gains intensity at low temperature, consistent with a change in valence state across the region below $T_\nu \approx 60$ K. To further confirm the intermediate valence scenario at low temperature, in Fig. 5c we show x-ray photoemission spectroscopy measurements at T = 12 K, which also show unambiguously the existence of the Sm$^{2+}$ state using an incident energy of 1400 eV, similar to the analysis of SmOs$_4$Sb$_{12}$ data carried out by Yamasaki et al.[29].

The energy difference observed between the Sm$^{2+}$ component and the Fermi energy $E_F$ observed in our Res-ARPES spectra raises the interesting possibility that some of the atomic multiplets of Sm play a role in the efficiency of the microscopic $f$ electron delocalization mechanism in 115 materials. While the appearance of the Sm$^{2+}$ level near $E_B = 0.4$ eV, marked by a dashed green line in Fig. 5b, might be explained by a surface-core shift similar to SmSn$_3$[31] and SmRh$_2$Si$_2$[33], we also cannot discount the possibility that a multiplet level of Sm that is not the ground state participates in the Abrikosov-Suhl resonance process. In the Kondo effect, the width of the Kondo resonance is an indicator of the separation between the Fermi energy and the localized magnetic state. In CeCoIn$_5$, CeRhIn$_5$, and CeIrIn$_5$, the width of the resonance indicates the Ce states to be less than 5 meV from $E_F$[39], while in SmB$_6$, the Sm$^{2+}$ states were shown to be 16 meV below $E_F$[54]. These results suggest that the width of the Kondo resonance in rare earth intermetallics is of order meV, far smaller than the separation of ~6 eV in SmCoIn$_5$ would allow for hybridization to proceed through the ground state of Sm$^{3+}$. However, the Coulomb repulsion between mutual $f$ electrons in Sm$^{3+}$ distribute the $\binom{14}{5} = 2002$ $f$ levels over more than 16 eV, as diagrammed in Fig. 2d, and the distribution of $\binom{14}{6} = 3003$ Sm$^{2+}$ multiplet states is similarly large. This fact suggests that a multiplet level of Sm could be found within 10–15 meV of $E_F$ and therefore have a much larger Kondo hybridization matrix element compared to the actual crystal field ground state wavefunction found in this work. This opens the intriguing possibility of multiplet structure participating in the Kondo effect at $E_F$.

Multiplet-selective hybridization between rare earths and metallic bands has been observed in other materials. In EuNi$_2$P$_2$, multiplets of the final-state $4f^6$ configuration were directly observed to hybridize selectively with the valence band structure[55]. In SmB$_6$, many-body correlations can explain the formation of the indirect

band gap that leads to a topological insulating state, with evidence for formation of isolated multiplet levels in one part of the spectral function and a single quasiparticle band in a different part[15]. In $SmSn_3$, high-resolution Res-ARPES combined with DFT predicted a shift of 0.13 eV of the ground multiplet with respect to $E_F$ that positions a higher multiplet directly at $E_F$[31]; however, this result was also dependent on some particular details of the calculations[33]. Perhaps most similarly to our work, a recent inelastic x-ray scattering experiment, combined with DMFT, suggested that the Kondo interaction proceeds in $CePd_3$ by shifting part of the rare earth spectral weight across the Fermi energy[56]. Therefore, multiplet hybridization effects may be important for the electronic structure and Kondo interaction, and our observation of a shift of the $Sm^{2+}$ state in $SmCoIn_5$ suggests that this possibility should be further explored. Given the large number of Sm-based compounds with heavy fermion and/or intermediate valence characteristics[37], it is likely that multiplet effects play a more frequent role than has been previously reported.

## Conclusions

In summary, we identified thermal isolation of the ground state crystal field doublet as a driving mechanism for $f$ electron delocalization in $SmCoIn_5$, a close relative of the unconventional superconductor $CeCoIn_5$. At low temperatures, the superposition of Sm valence states resembles a Kondo scenario that also hosts charge fluctuations between the $Sm^{3+}$ and $Sm^{2+}$ configurations. The delocalization begins when the temperature is low enough that the shape of the single-ion Sm wavefunction becomes more box-like compared to a filled shell containing thermally populated excited states. At the same temperature, we begin to see magnetism appear in the bulk properties, suggesting a form of the hybridization that involves a magnetic degree of freedom. In this way, we connect delocalized itinerant band states to the single-ion properties of $Sm^{3+}$ through a generalization of the Kondo effect to the multi-$f$-electron systems. Finally, since the single-ion properties and crystal field ground state of $CeCoIn_5$ are similar to those we found in $SmCoIn_5$, similar charge fluctuations may also be present in the superconducting dome of $CeCoIn_5$. However, in $SmCoIn_5$, the ground state is antiferromagnetically ordered[38], indicating that the Kondo screening is not sufficient to form a Kondo lattice, and we do not observe any signatures of Kondo coherence. Our results therefore show that $SmCoIn_5$ is in a hybrid region close to the quantum critical point containing aspects from both sides, where the microscopic ingredients for $f$ delocalization are present, but the magnetic RKKY exchange nevertheless becomes dominant.

## Methods

**Sample growth**. Single crystals of $SmCoIn_5$ and $SmIrIn_5$ were grown by a molten In flux method. For $SmCoIn_5$, the starting materials were Sm (99%, Goodfellow Cambridge Ltd.), Co (99.9+ %, Alfa Aesar), and In (99.9999%, Alfa Aesar). Sm (1.078 g, 0.717 mmol) and Co (0.4223 g, 0.717 mmol) were reacted together by means of arc melting method with negligible weight loss (less than 0.15%). In a helium-filled glovebox, the resulting materials were placed into a 5 ml alumina Canfield crucible[57] and mixed with In (31.999 g, 27.869 mmol). The crucible set was placed in a quartz ampule, evacuated, backfilled with ~100 mbar of Ar, and sealed in a quartz ampoule. The specimen was heated up to 1150 °C, with a rate 400 °C/h, and annealed at that temperature for 1 h. Annealing at 1150 °C was assisted by motorized rotation of the ampoule for better homogenization. Subsequently, the sample was cooled down to 800 °C with a rate of 400 °C/h, and further cooled down to 350 °C with a rate of 1 °C/h. After the growth step, the excess In flux was separated from the single crystals using a centrifuge. For $SmIrIn_5$, the starting materials

were 0.820 g (0.545 mmol) of Sm rod; 1.048 g (0.545 mmol) of Ir powder (2N5+; chemPUR), and 33.531 g (29.204 mmol) of In. The starting materials were placed directly into the crucible without the arc melting step. During the growth, the specimen was held at 1150 °C for 3 h, and cooled to 450 °C with a rate of 5 °C/h for the centrifugation. For both materials, the remaining In flux was dissolved in aqueous (37%) HCl. The sample composition was confirmed by x-ray fluorescence spectroscopy and powder x-ray diffraction. The $SmCoIn_5$ samples exhibited a plate-like habit corresponding to the basal plane of the crystal, while $SmIrIn_5$ crystals appeared to be more block-like.

**XAS measurements**. X-ray absorption spectroscopy (XAS) experiments were carried out at the X-Treme beamline at the Swiss Light Source, PSI, Switzerland[58]. Samples were mounted on a copper plate using silver paint, including drops of silver paint connecting the copper to the top surface of the sample, ensuring excellent electrical contact with the area exposed to the beam. To ensure consistent sample preparation, we used three methods and tested all of them for differences: first, we prepared as-grown samples with no surface preparation; second, we cleaved samples before introducing the samples into the vacuum chamber; third, we cleaved the samples in a helium glove box before transferring them into the experimental chamber. We collected XAS scans on all three samples and searched for changes in the spectra under these conditions, or a reduction of the intensity, but we found that all three were equivalent; therefore, we rule out any surface-specific effects in our data. The samples were mounted vertically on a rotating cold finger with the horizontal plane corresponding to the $(100) \times (001)$ plane of the crystal axes. To measure the x-ray linear dichroism (XLD), the samples were rotated into grazing incidence so that the beam direction changed by 60 degrees from the (001) axis toward the (100) axis, which coincidentally is very close (about 1 degree) to the (101) direction in reciprocal space. A diagram of the experimental configuration of the XAS experiments is shown in the Supplementary Fig. 5. In this way, horizontally polarized light probes a component in the (001) direction which is the sine of the rotation angle. Additional experiments of the x-ray magnetic circular dichroism (XMCD) were carried out at the same grazing incidence, with the magnetic field of up to 6.8 T applied in the direction of the beam. To analyze the XAS experimental data, the scans were normalized such that the height of the largest edge (M5 or L3), averaged between the polarizations, was 1 relative to the value of the scan before the pre-edge.

**Single-ion multiplet calculations**. Full multiplet calculations of Sm, including magnetic susceptibility and XLD patterns at the Sm $M_{4,5}$ edges, were performed with the software package QUANTY[42], similar to the calculations presented by Sundermann et al.[59]. The atomic parameters were taken from the Crispy interface for QUANTY[60] which reproduces the values from the Cowan code, and the values were reduced to 80% in order to reproduce our experimental results, which is a typical value[59]. The values of spin-orbit coupling were found to be 180 meV for the Sm $4f$ states, 200 meV for the Sm $4f$ states in the core-hole state, and 10.51 eV for the Sm $3d$ states. Gaussian broadening of 0.05 eV was applied to the spectra to simulate the experimental conditions.

**Magnetization measurements**. Bulk magnetization measurements were carried out in a Quantum Design MPMS3 vibrating sample magnetometer (VSM). For measurements with the field in the basal plane of the single crystal, the sample was attached to a quartz sample paddle with GE varnish and wrapped with teflon tape. For measurements with the field along the (001) axis, the

samples were mounted between two quartz cylinders inside a brass straw.

**Inelastic neutron scattering measurements**. Inelastic neutron scattering was carried out with a 1.8 g powder sample of $SmIrIn_5$ at the EIGER thermal triple-axis spectrometer, PSI, Switzerland. Because of the extremely strong thermal neutron absorption resonance in natural Sm, we could not mount the samples in a standard powder can. Therefore, we mixed the powder with several drops of Cytop hydrogen-free glue and spread it as evenly as possible on a 9 cm × 4 cm aluminum plate of thickness 0.5 mm. The aluminum plate was mounted in the spectrometer with its long axis at a fixed angle to the incoming beam, as the outgoing beam angle was varied in the process of conducting energy scans at constant scattering wavevector Q. For each energy scan at constant Q, we fixed the sample angle at a single optimal position for the entire scan, to avoid having the long direction of the plate coincident with either the incoming and outgoing beam, which would lead to very large neutron absorption.

**X-ray diffraction measurements**. Temperature-dependent lattice parameter measurements were carried out in the cryostat at the MS beamline at the Swiss Light Source, PSI, Switzerland. Powdered $SmCoIn_5$ was prepared and diluted with cornstarch by about 50% by volume in order to ensure constant illumination, then placed into a quartz capillary of diameter 0.3 mm. We used a beam energy $E = 22$ keV ($\lambda = 0.563564$ Å), which is below the In absorption edge near 27 keV. The wavelength was calibrated using a silicon standard, while starting u,v,w parameters used in the refinements were determined using the $LaB_6$ NIST660A standard. The temperature in the cryostat was varied between 4 and 300 K. Refinements were carried out using the Fullprof software package and found to agree with the expected lattice structure. In Fullprof, an absorption correction $\mu R = 0.9$ was applied, and we used $X = 0.47$ and $Y = 0.00017$. We found evidence for a secondary phase of $Sm_2CoIn_8$ which was 2.9% by volume, which we assume comes from small crystallites on the edges of the $SmCoIn_5$ single crystals (see Supplementary Note 5 and Supplementary Fig. 8).

**ARPES measurements**. Angle-resolved photoemission spectroscopy (ARPES) experiments were carried out at the soft-x-ray endstation[61] of the ADRESS beamline[62] at the Swiss Light Source, PSI, Switzerland. Single crystals were mounted using the natural basal plane on copper plates using high-strength H20E silver epoxy. Posts made of small stainless steel screws were attached onto the top surfaces of the samples with Torr-Seal epoxy. The samples were cleaved in ultrahigh vacuum of about $10^{-10}$ mbar. We used circularly polarized light with a beam spot of about $60 \times 100$ μm on the sample. We determined the coupling of beam energy to the out-of-plane $k_z$ direction by scanning the beam energy over $h\nu = 480–700$ eV, and we identified $h\nu = 602$ eV as a scan through the Γ point and $h\nu = 564$ eV as a scan through $Z$. Maps were then collected at these energies to determine the Fermi surface and valence band structure, where we found broad similarity to the published results of $CeCoIn_5$[5]. We also conducted XPS measurements (Fig. 4c) as well as resonant partial spectra at the Sm $M_{4,5}$ edges (Fig. 5b).

**DFT calculations**. All the density functional theory calculations were carried out using the Quantum ESPRESSO package[63–65]. Exchange and correlation effects were modeled using the PBEsol functional[66], augmented by a Hubbard U term to better describe the physics of the localized Sm 4$f$ electrons. The scalar-relativistic pseudopotentials are taken from the SSSP library[67]. To sample the Brillouin zone, we used a $5 \times 5 \times 5$ Monkhorst-Pack grid; a grid twice as fine was used for the calculation of the projected density of states. The wave-functions (charge densities and potentials) were expanded using a kinetic energy cutoff of 50 Ry (400 Ry). We used the experimental geometry[66,67] throughout this work. The values of U used in this work (U = 6.03 eV and U = 6.26 eV for $Sm^{3+}$ and $Sm^{2+}$ calculations, respectively) were computed fully ab-initio by using the linear response approach[68] as implemented by Timrov et al.[69,70]. In order to force the self-consistent-field calculations to converge to the $Sm^{3+}$ solution, we constrained the number of electrons in the Sm $f$ shell to be 5. The data used to produce the simulation results presented in this work are available at the Materials Cloud Archive[71].

**Transport measurements**. Transport measurements were performed in a Quantum Design PPMS equipped with a 9 T magnet and a sample rotator. The sample was mounted on a sapphire plate with GE varnish, and leads were painted on by hand in a Hall bar geometry with current always flowing along the (100) axis. Aluminum wires were then attached to the paint leads by wire bonding. For the Hall effect measurements, we oriented the magnetic field along the (001) direction, we used a spear-shaped single crystal which was longer along the (100) axis compared to the (010) axis, ensuring a nearly Hall-bar configuration.

## Data availability

All relevant data are available from the corresponding authors upon reasonable request. The data used to produce the DFT simulation results presented in this work are available at the Materials Cloud Archive[71].

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

## Acknowledgements
We are grateful for useful discussions with Andriy Nevidomskyy, Qimiao Si, Collin Broholm, Markus Müller, and Manfred Sigrist, and we thank Uwe Stuhr for assistance with planning the Eiger experiments, Romain Sibille for assistance with the MS experiments, and Jamie Massey for assistance with the VSM measurements. This research was supported by the Swiss National Science Foundation Project No. 200021_184983 (M.K.), and by the NCCR MARVEL, a National Centre of Competence in Research, funded by the Swiss National Science Foundation (grant number 205602) (N.C.). D.W.T. and N.K. acknowledge funding from the European Union's Horizon 2020 research and innovation programme under the Marie Skłodowska-Curie grant agreement, No 884104 (PSI-FELLOW-III-3i) (D.W.T.) and No. 884104 (PSI-FELLOW-III-2i) (N.K.). F.A. acknowledges support by Swiss National Science Foundation Project No. 206312019002.

## Author contributions
D.W.T. and M.K. conceived the work and guided the project. D.J.G. and E.P. prepared SmCoIn$_5$ and SmIrIn$_5$ samples and characterized their composition and quality. N.C. performed density functional theory calculations. D.W.T. and N.K. prepared transport experiments, and N.K. and Y.S. conducted the transport experiments and analyzed the

results. D.W.T. and C.P. prepared and conducted XAS experiments with input from N.K. and Y.S., and D.W.T. analyzed the XAS data and conducted the full-multiplet simulations. D.W.T., F.A., and V.N.S. prepared and conducted ARPES experiments and analyzed the data. D.W.T. and A.C. prepared and conducted powder x-ray experiments and D.W.T. carried out the Rietveld refinement analysis with input from D.J.G. D.W.T. and T.F. prepared and conducted inelastic neutron scattering experiments. The paper was written by D.W.T. and M.K. All authors provided comments on the manuscript.

## Competing interests

The authors declare no competing interests.
