## [Peer Review File · Communications Physics]

Reviewers' comments:

Reviewer #1 (Remarks to the Author):

Nature Communications Physics article COMMSPHYS-23-0294-T
Charge fluctuations in the intermediate-valence ground state of SmCoIn₅ (Tam et al.)

Tam et al. present a thorough characterization of Kondo hybridization phenomena in the intermediate valence (IV) compound SmCoIn₅. The authors used a suitable range of complementary experimental techniques and connected their results using well-established computational methods. The key results are

- The determination of the crystal field structure and ground state of SmCoIn₅
- The observation of a valence crossover and the identification of its mechanism

IV materials provide exciting opportunities to directly observe the incipient hybridization between local and itinerant electronic degrees of freedom, including both charge (orbital) and magnetic correlations. Materials like SmCoIn₅ give experimental access to the crucial ingredients of the local-itinerant dichotomy at the heart of quantum critical phenomena. The universality of these effects makes such insights (e.g. on the mechanisms of hybridization) relevant to a large proportion of quantum materials research.

The absence of an order parameter in the valence fluctuating state makes it difficult to characterize these materials by a single experimental technique. The present study is exemplary in that it showcases how results from x-ray (and neutron) spectroscopic methods can be connected to bulk magnetometry, diffraction, and charge transport to arrive at a consistent picture and convincing conclusion.

While the archetypal IV materials are Ce or Yb intermetallics, there is an added interest in Sm and Eu, where the complexity cannot be reduced to a single electron or hole. In the present manuscript, the salient questions of the field (e.g. the origin of emerging energy scales), are introduced in a didactic introduction section that will also be accessible to a non-specialist audience.

The paper is written in clear, grammatically correct and stylistically appealing English. The small number of typos gives the impression that the manuscript was prepared with great care. It is also well structured, and a clear message is attributable to every paragraph. In many instances, the upshot of a longer discussion is briefly summarized at the end of the corresponding section – I found that this made the paper particularly readable.

The illustrations and schematics in Figs. 1 and 2 provide a useful guide to the manuscript and summarize the main results at a glance. Similarly, the data in Figs. 3-5 is presented with helpful annotations. Considering the variety of experimental probes used in this study, I felt that the paper read remarkably coherent.

The data and analysis appear sound and the key argument (that the valence crossover is determined by the thermal population of CEF states) is well supported by the data. In the Materials and Methods section and in the Supplementary Information file, all experimental, technical and computational aspects of the study are described at a commendable level of detail. This would certainly enable researchers in the field to attempt to corroborate the results.

I am convinced that the paper will be of interest to any researcher in the intermediate valence community, and also of relevance to the wider fields of lanthanides and actinides electronic structure, including quantum criticality in heavy fermion compounds, and unconventional superconductivity in general. I would definitely recommend publication. I only have few remarks and questions that should be addressed:

- Caption to Fig. 2: It is a bit confusing that the level schemes and Stevens operators of the Ir compounds (which are shown for reference in the figure) are not addressed in the caption.
- Caption to Fig. 2: I was also slightly confused by the notation of the spin-orbit-interactions – what are the different conventions for ξ vs. Δ ? Only one version should be used (for Ce, $\xi=87$ meV is given in the caption, but $\Delta=300$ meV is stated in the figure).
- Fig. 3(a): In print, the XLD spectra will not be distinguishable, which may make the legend confusing. Maybe one can avoid this by drawing one of the spectra with a dashed line.
- Fig. 3(b): I would be interested in how exactly the Stevens parameters were determined. Was the data iteratively fitted with the results of the Quanta algorithm? Simply put: How were the error bars shown in Fig. 2(h) extracted? (this could also be addressed in the SI)
- Table I (typo): remove duplicate "TABLE I. Table 1."
- Caption to Table I: It seems the value for A_4^4 (50 meV) deviates from the one shown in Fig. 2(h) (40 meV)? If I did not overlook this, the Stevens parameters and their uncertainties are not explicitly quoted in the main text. It would be good to add this information (cf. comment above).
- Fig. 4(a,b,c): Regarding the magnetometry data and its Quanta model: If I understand correctly, the calculations of the magnetic susceptibility and the level population (inset) are obtained from the same code. So their comparison amounts only to an illustration of the model itself, regardless of the susceptibility data. It seems a strange coincidence that the incipient magnetization of Co 3d bands (inferred from XMCD) makes up precisely for the difference to a continuous CW fit (with a single CW component down to below T_v). I would be very interested if the authors could confirm/elaborate on this point. What temperature range was taken into account for the CW fit?
- Fig. 5(c): Is temperature-dependent data of the XPS data available? Would this also allow tracking the valence changes? Is it possible to make a quantitative estimate of the Sm valence (f-shell occupation) based on the experimental data?
- Line 392: (typo) insert "at" ["(...) states at the Fermi level"]
- Line 460: (typo) binomial quotient $\binom{14}{6}=3003$

Reviewer #2 (Remarks to the Author):

In their manuscript the authors report results from X-ray and neutron spectroscopy as well as thermodynamic measurements on SmCoIn5. From these results they deduce a crystal electric field scheme for the Sm ions with excited levels at about 11-14 meV. This scheme is then connected to a cross over that the authors observed at around 60 Kelvin in several physical properties and the X-ray spectroscopy. As the main conclusion of their paper, the authors argue for a partial delocalisation of Sm 4f states below 60K that is related to a change in shape of the Sm 4f wave function from almost spherical to more box-like. This box-like shape is seen as favorable for hybridisation and thus as giving rise to 4f electron delocalisation below 60K.

I find the presented experimental data of very good quality and extremely interesting, given the current lack of data in the literature on SmCoIn5. I would therefore like to recommend this manuscript for publication and I am sure it will be received with a lot of interest in the 4f electron community.

The presented interpretation of the experimental data is less convincing to me. While the scenario developed by the authors may well be correct, I believe that their data also allows for other interpretations. In particular, I think the authors should be more critical regarding the sensitivity to surface effects in the soft X-ray XAS and particularly Res-ARPES data which most of their conclusions are based on. I would therefore encourage a revision of the manuscript before publication. Below I can give a couple of points which I think should be addressed in some form.

1. A good part of the interpretation hinges on the crystal field level scheme established from the XAS data in Fig. 3. Electron yield XAS is sensitive to surface relaxation effects that can lead to a different 4f level scheme close to the surface as compared to the bulk. How can the authors be sure that the measured XLD data is not affected by surface relaxation effects and fully reflecting bulk properties.

2. Can the authors show the XAS spectra, from which the XLD in Fig. 3 was derived, as a function of temperature. Do the authors see a growing Sm^{2+} below T_v and no Sm^{2+} above T_v in agreement with their model and consistent with the temperature dependence observed in the photoemission?

3. If one accepts that the XLD data is purely bulk, how can the authors be sure that there is a unique solution for the CEF to explain the observed XLD vs T?

3. Similar to the XAS data, can the authors comment on possible effects of large surface sensitivity in the PES data? Hybridisation phenomena, CEF effects and even 4f ionisation states at the surface can be different to that in the bulk. Different surface terminations can completely change photoemission spectra even in the soft X-ray range. How can the authors be sure that their spectra show bulk properties, and not a mixture of surface and bulk properties?

4. Can the excitation at 16 meV in the SmIrIn_5 INS also be due to a single CEF level with a second level either close to zero or at higher energy. If the analogy to the CeMIn_5 ($M=\text{Co,Rh,Ir}$) holds, one would expect a smaller splitting between the two G7 levels than between the excited G7 and the G6. From a visual inspection the width of the 16 meV peak seems compatible with one single excitation and the line shapes used to fit this peak seem narrower than those fitting the other peaks. Can the authors comment? Do the presented INS data exclude a quasi quartet ground-state or another CEF 30 meV?

5. I can follow the authors' argument, that the crystal field in SmCoIn_5 and SmIrIn_5 is likely not very different, based on the very similar CEF schemes in the CeMIn_5 systems. As INS on SmCoIn_5 was not possible, the SmIrIn_5 experiment seems meaningful also to discuss SmCoIn_5 . However, in order to connect with and give confidence in the SmCoIn_5 XLD results, XLD measurements on SmIrIn_5 would have been very important. Did the authors try to do such measurements?

6. The susceptibility data seems very poorly fit but the proposed CEF model, not only below but also above 100K. Can the authors show the inverse susceptibility and the fit with their CEF model in order to better judge the high-T region where the CEF model should fit the data nicely.

7. In the low temperature region, the proposed CEF model deviates dramatically and qualitatively from the data. The susceptibility data reported here is very similar to that seen in CeMIn_5 and there a good correspondence between the measured susceptibility and its in-plane to out of plane anisotropy with the proposed CEF models was found. The authors invoke an emerging Co magnetic moment at low T due to 4f transfer into Co d states as the reason. I am not convinced that this is actually explain the observed deviation between their model and the data.

a) Could the temperature dependence of the XMCD detected Co moments also be explained as the response of constant, paramagnetic Co moments at fixed field vs T. Have the authors measured $M(H;T)$ to be sure that moments are saturated at 7T at least between 20 and 70K?

b) The XMCD data was measured at 7T, but the susceptibility at 0.5 T. It is hard to relate the two measurements without $M(H)$ information. Similar to the XLD there's also the problem of surface vs bulk sensitivity when comparing these to techniques. Can the authors comment please.

c) What is the uncertainty on the extracted Co moments? The XMCD does not seem to vanish above the Co edge in SFig. 6.

d) Is the size of the detected Co moments (0.4 μ_B at 30K) compatible with the missing 0.4 to 0.5 emu/mol in the susceptibility? The estimated Co moments seem to large

e) How do the authors explain the directional dependence in Fig. 4c. Note that the field in the XMCD measurements was along 001 while the data points follow the 110 rather than the 001 susceptibility data.

f) The same poor fit to the low T region is also seen for SmIrIn5. Do the authors invoke the same explanation for the poor correspondence between their CEF model and the data in the Ir case, namely that 4f states start to delocalise into Ir 5d states and cause a finite 5d magnetic moment to just the right amount to recover very conventional Curie-Weiss behaviour in macroscopic measurements?

7. I find the paragraph starting from line 438 several times, especially the part starting line 454, difficult to understand. Can the authors rephrase the main message of this section please.

8. Can the observed deviation from linear behaviour in the Hall effect also be explained within a CEF model without invoking hybridization (see e.g. J. Low Temp. Phys. 22, 665, 1976)?

In summary, I think the authors could be more cautious and critical regarding their interpretation of the data. The CEF model anisotropy determined from the surface sensitive XLD does not seem supported by the bulk susceptibility measurements. The INS results on SmIrIn5 seem also be compatible with a single CEF excitation in the studied energy interval. The only clear sign of hybridization seems to come from the photoemission data, which could be sensitive to surface effects. Maybe the ideas put forward by the authors are correct, but I think the shown data also allows for other interpretations and that this should be reflected in the discussion in the manuscript.

minor comment:

How do the authors interpret the dispersing excitation in the INS data?

Reviewer #3 (Remarks to the Author):

The authors explored the Sm ion's charge fluctuation in SmCoIn5 and found evidence for the delocalization of Sm 4f electrons below 60 K. A lot of characterizations were conducted, making the analyses complete and results convincing. I recommend it publish in CommsPhys. Below are my comments/questions to the authors.

1) In Figure 1, (a) and (b) should be marked on the left and right panels of the figure, respectively.

2) Line 133, "n_f" in the parentheses should be " Δn_f ".

3) Lines 294-297: I guess the authors actually mean "...deviate from the experimental results"?

4) The authors show Res-ARPES and XAS of SmCoIn5 below 60 K. In order to support their claim that the Sm is mainly in its trivalent state above 60 K, it would be better to give the relevant data above

60 K. Do they just show a single peak for Sm³⁺?

5) It was claimed that there is an "upturn" in resistivity vs temperature curve below 60 K. However, it does not look like an upturn. It is just gradually deviating from $R \sim T$, which is quite common in even nonmagnetic metals because at lower temperatures resistivity is dominated by the T^5 term of the phonon scattering. Not sure if it makes sense to divide $R(T)$ into two regions by 60 K.

Charge fluctuations in the intermediate-valence ground state of SmCoIn_5

Reply to referee comments received April 28.

David W. Tam, Nicola Colonna, Neeraj Kumar, Cinthia Piamonteze, Fatima Alarab, Vladimir N. Strocov, Antonio Cervellino, Tom Fennell, Dariusz Jakub Gawryluk, Ekaterina Pomjakushina, Y. Soh, and Michel Kenzelmann
(Dated: May 27, 2023)

To the editor and referees:

We are very pleased to have received all of your comments and look forward to the opportunity to address each of the points in detail.

REPLY TO REFEREE 1

Tam et al. present a thorough characterization of Kondo hybridization phenomena in the intermediate valence (IV) compound SmCoIn_5 . The authors of used a suitable range of complementary experimental techniques and connected their results using well-established computational methods. The key results are

- The determination of the crystal field structure and ground state of SmCoIn_5*
- The observation of a valence crossover and the identification of its mechanism*

IV materials provide exciting opportunities to directly observe the incipient hybridization between local and itinerant electronic degrees of freedom, including both charge (orbital) and magnetic correlations. Materials like SmCoIn_5 give experimental access to the crucial ingredients of the local-itinerant dichotomy at the heart of quantum critical phenomena. The universality of these effects makes such insights (e.g. on the mechanisms of hybridization) relevant to a large proportion of quantum materials research.

The absence of an order parameter in the valence fluctuating state makes it difficult to characterize these materials by a single experimental technique. The present study is exemplary in that it showcases how results from x-ray (and neutron) spectroscopic methods can be connected to bulk magnetometry, diffraction, and charge transport to arrive at a consistent picture and convincing conclusion.

While the archetypal IV materials are Ce or Yb intermetallics, there is an added interest in Sm and Eu, where the complexity cannot be reduced to a single electron or hole. In the present manuscript, the salient questions of the field (e.g. the origin of emerging energy scales), are introduced in a didactic introduction section that will also be accessible to a non-specialist audience.

The paper is written in clear, grammatically correct and stylistically appealing English. The small number of typos gives the impression that the manuscript was prepared with great care. It is also well structured, and a clear message is attributable to every paragraph. In many instances, the upshot of a longer discussion is briefly summarized at the end of the corresponding section – I found that this made the paper particularly readable.

The illustrations and schematics in Figs. 1 and 2 provide a useful guide to the manuscript and summarize the main results at a glance. Similarly, the data in Figs. 3-5 is presented with helpful annotations. Considering the variety of experimental probes used in this study, I felt that the paper read remarkably coherent.

The data and analysis appear sound and the key argument (that the valence crossover is determined by the thermal population of CEF states) is well supported by the data. In the Materials and Methods section and in the Supplementary Information file, all experimental, technical and computational aspects of the study are a described at a commendable level of detail. This would certainly enable researchers in the field to attempt to corroborate the results.

I am convinced that the paper will be of interest to any researcher in the intermediate valence community, and also of relevance to the wider fields of lanthanides and actinides electronic structure, including quantum criticality in heavy fermion compounds, and unconventional superconductivity in general. I would definitely recommend publication.

We are very appreciative of the referee's supportive and detailed comments.

- Caption to Fig. 2: It is a bit confusing that the level schemes and Stevens operators of the Ir compounds (which are shown for reference in the figure) are not addressed in the caption.*

47 • *Caption to Fig. 2: I was also slightly confused by the notation of the spin-orbit-interactions – what are the different*
 48 *conventions for ξ vs. Δ ? Only one version should be used (for Ce, $\xi=87$ meV is given in the caption, but $\Delta=300$ meV is*
 49 *stated in the figure).*

50 • *Fig. 3(a): In print, the XLD spectra will not be distinguishable, which may make the legend confusing. Maybe one can avoid*
 this by drawing one of the spectra with a dashed line.

51
 52 We are grateful for these comments. In the revised manuscript, we have added the Stevens operator parameters of both
 53 SmCoIn_5 and SmIrIn_5 to the caption of Fig. 1, and we added dashing to one of the curves in the XAS plot in Fig. 3a. The
 54 tabulated values of spin-orbit interaction ξ come from the Crispy interface (Ref. 62 in the paper), and are 87 meV for Ce, and
 55 102 meV for Sm, which are the parameters used in the full-multiplet calculations. These values reflect differences in the radial
 56 wavefunction $R(r)$, and enter the Hamiltonian through the spin-orbit term $\xi \mathbf{L} \cdot \mathbf{S}$. The splitting between J multiplets, Δ , is further
 57 scaled due to the various quantum numbers involved. Going from Ce to Sm, it is typical to find a reduction in the multiplet
 58 separation by about a factor 2, as can be seen in a variety of books about trivalent rare earths. One publication with such a figure
 59 is: D’Aléo, A., Pointillart, F., Ouahab, L., Andraud, C. & Maury, O. Charge transfer excited states sensitization of lanthanide
 60 emitting from the visible to the near-infra-red. *Coordination Chemistry Reviews* 256, 1604–1620 (2012).

61 In general, we chose to include the actual expected splitting Δ in the level diagram, whereas for cases where we describe the
 62 parameters used as input to the calculations, we state the bare values ξ . We also added a sentence in the caption of Fig. 2 to
 63 clarify this:

64 The CeCoIn_5 calculations in (a-c) were performed for the $4f^1$ configuration using the crystal field parameters reported in Ref.
 65 [39] and spin-orbit interaction $\xi_{\text{SO}} = 87$ meV, which leads to a splitting between the $J = 5/2$ and $J = 7/2$ spin-orbit manifolds of
 66 $\Delta \approx 300$ meV. The SmCoIn_5 calculations in (d-g) were performed using the parameters found in this work and with $\xi_{\text{SO}} \approx 102$
 67 meV, which leads to a lower $\Delta \approx 180$ meV.

68 • *Fig. 3(b): I would be interested in how exactly the Stevens parameters were determined. Was the data iteratively fitted with*
 69 *the results of the Quanty algorithm? Simply put: How were the error bars shown in Fig. 2(h) extracted? (this could also be*
 addressed in the SI)

70
 71 The Stevens parameters were found by exhaustively searching through the parameter space of three operators (A20, A40,
 72 A44), and computing the temperature dependence of the XLD for every set and comparing it to the experimental data. Since
 73 each calculation takes two hours (it is difficult to diagonalize the core-hole excited state), the choice of how to explore the space
 74 was done by hand. In general, it is very difficult to come up with a level scheme that matches the XLD pattern at all temperatures.
 75 For instance, in order to capture the inversion of the XLD pattern near $T=60$, it is important to look at level schemes for the
 76 ground multiplet which contain at least one level near $E=10$ -20 meV, which already eliminates a lot of the parameter space. After
 77 determining the CEF levels of SmIrIn_5 using INS, we further reduced the parameter space of interest using the metric of a level
 78 separated by 15-20 meV and another level close to one of those two energies, which is consistent with our final interpretation.
 79 Moreover, this approach does not rule out other areas of the parameter space because we found no other solutions that contained
 80 a suitable agreement as well as an inversion near $T=60$ K.

81 We show several examples of the calculated XLD patterns, and their comparison to our data, in Fig. 1. In the top row, we
 82 show the experimental data and our model calculations for all temperatures plotted all on the same set of axes. In the rows below,
 83 we show results of calculations using other choices of Stevens parameters. Many sets of parameters lead to obviously wrong
 84 choices. One key feature in the experimental data is an inversion of the pattern near $T=60$ K, and we used this to exclude many
 85 choices of parameters. For example, for the set indicated by Stevens parameters (-6,170,35), the crossover temperature shifts
 86 down to about $T=50$, which is less likely than the parameter set we chose (-4,170,50), although we note that this solution would
 87 still be contained within the error bars presented in Fig. 2 of the paper.

88 We also used the calculated magnetic susceptibility to narrow the choices of Stevens parameters. In Fig. 2, we show that for
 89 a model using parameters (0,180,30), the trend in susceptibility has a slope in the range $T=100$ -300 K (smooth lines) that does
 90 not match the curvature of the measured data (noisy lines). Since the susceptibility shown in the paper matches the XLD and
 91 the magnetic susceptibility, is consistent with the INS data on SmIrIn_5 , and is not far away from the values of A40 and A44
 92 compared to CeCoIn_5 and CeIrIn_5 as shown in Fig. 2 of the paper, we conclude that the evidence overwhelmingly supports our
 93 determination of the crystal field parameters.

94 To answer the question about error bars, these are not strictly defined because we could not come up with a consistent measure
 95 of the goodness of fit. Comparing the XLD patterns from different calculations, we found that the dipole term A20 is more tightly
 96 constrained before the fit becomes poor, while there is about 20 meV of room to vary both A40 and A44. For SmIrIn_5 , we could
 97 only estimate the error based on the range of parameters leading to the level spacing found by INS and matches the curvature of
 98 the magnetic susceptibility, therefore the error bars are larger for that material.

99 To clarify these details in the paper, we have changed part of the caption of Fig. 2 to be more descriptive:

100 (h) Stevens A parameters for the crystal field potential used in the calculations, in meV: (-4, 170, 50) for SmCoIn_5 and (2,
 101 240, 80) for SmIrIn_5 , which were found via an exhaustive search in the space of A_2^0 , A_4^0 , and A_4^4 . For SmCoIn_5 , the error bars
 102 are chosen by hand to account for reasonable variation in the parameters that still remains a close match to the experimental data
 103 (XLD and magnetic susceptibility) and reproduce the temperature ($T_v = 60$ K) at which the XLD pattern inverts. For SmIrIn_5 ,
 104 the error bars were chosen to match INS and magnetic susceptibility.

- 105 • *Table I (typo): remove duplicate “TABLE I. Table 1.”*
 106 • *Caption to Table I: It seems the value for A44 (50 meV) deviates from the one shown in Fig. 2(h) (40 meV)? If I did not*
 107 *overlook this, the Stevens parameters and their uncertainties are not explicitly quoted in the main text. It would be good to add*
this information (cf. comment above).

108
 109 Thank you for these corrections, we have made both revisions.

- 110 • *Fig. 4(a,b,c): Regarding the magnetometry data and its Quany model: If I understand correctly, the calculations of the*
 111 *magnetic susceptibility and the level population (inset) are obtained from the same code. So their comparison amounts only*
 112 *to an illustration of the model itself, regardless of the susceptibility data. It seems a strange coincidence that the incipient*
 113 *magnetization of Co 3d bands (inferred from XMCD) makes up precisely for the difference to a continuous CW fit (with a single*
 114 *CW component down to below T_v). I would be very interested if the authors could confirm/elaborate on this point. What*
temperature range was taken into account for the CW fit?

115
 116 Thank you for considering this point. It is correct that the magnetization is calculated from the Quany model used to fit the
 117 XLD patterns.

118 We were also initially surprised by this “coincidental” result. Our interpretation is that the calculated susceptibility should
 119 in principle be only true for strongly localized moments, such as in insulators. When delocalization and hybridization with the
 120 metallic bands occurs, the magnetic moments of the multiplet states of Sm which favor hybridization could provide the missing
 121 magnetic susceptibility.

122 To illustrate this picture another way, consider that the three wavefunctions (of the three Kramers doublets) together make up a
 123 rotationally-symmetric shell. In the ground state, the localized Sm electrons adopt only the Γ_7 wavefunction, but the delocalized
 124 Sm component could in principle take some other shape that minimizes the energy of the hybridized electronic structure. The
 125 fact that this actually seems to occur in SmCoIn₅ is something we find remarkable. However, we decided not to make it a
 126 primary goal of this paper.

127 We also note that we tried to find a crystal field solution giving a different ground state (with no competing level at $E \sim 10$
 128 meV), which would then exhibit a Curie-Weiss-like susceptibility that reproduces the M(T) data. In all cases, we found that
 129 these models did not exhibit an inversion of the XLD pattern. Therefore, having found a good match for the high-T part of the
 130 susceptibility, which also matches the XLD pattern at all temperatures, is evidence that the delocalized moments make up the
 131 remainder of the experimental susceptibility below $T = 60$.

- 132 • *Fig. 5(c): Is temperature-dependent data of the XPS data available? Would this also allow tracking the valence changes?*
Is it possible to make a quantitative estimate of the Sm valence (f-shell occupation) based on the experimental data?

133
 134 Unfortunately, we did not have time to collect temperature dependence of the XPS data. During the beam time, we chose
 135 to prioritize obtaining good statistics with Res-ARPES. We agree that this measurement should make a very nice piece of
 136 experimental evidence.

- *Line 392: (typo) insert “at” [“(…) states at the Fermi level”]* • *Line 460: (typo) binomial quotient $(14-6)=3003$*

137
 138 Thank you for pointing out these typos, we have fixed them.

139 REPLY TO REFEREE 2

140 *In their manuscript the authors report results from X-ray and neutron spectroscopy as well as thermodynamic measurements*
 141 *on SmCoIn₅. From these results they deduce a crystal electric field scheme for the Sm ions with excited levels at about 11-14*
 142 *meV. This scheme is then connected to a cross over that the authors observed at around 60 Kelvin in several physical properties*
 143 *and the X-ray spectroscopy. As the main conclusion of their paper, the authors argue for a partial delocalisation of Sm 4f states*
 144 *below 60K that is related to a change in shape of the Sm 4f wave function from almost spherical to more box-like. This box-like*
 145 *shape is seen as favorable for hybridisation and thus as giving rise to 4f electron delocalisation below 60K.*

146 *I find the presented experimental data of very good quality and extremely interesting, given the current lack of data in the*
 147 *literature on SmCoIn₅. I would therefore like to recommend this manuscript for publication and I am sure it will be received*

148 *with a lot of interest in the 4f electron community.*

149 *The presented interpretation of the experimental data is less convincing to me. While the scenario developed by the authors*
 150 *may well be correct, I believe that their data also allows for other interpretations. In particular, I think the authors should be a*
 151 *more critical regarding the sensitivity to surface effects in the soft X-ray XAS and particularly Res-ARPES data which most of*
 152 *their conclusions are based on. I would therefore encourage a revision of the manuscript before publication. Below I can give a*
couple of points which I think should be addressed in some form.

153

154 We are very grateful to this referee for the enthusiasm and detailed comments.

155 *1. A good part of the interpretation hinges on the crystal field level scheme established from the XAS data in Fig. 3. Electron*
 156 *yield XAS is sensitive to surface relaxation effects that can lead to a different 4f level scheme close to the surface as compared*
 157 *to the bulk. How can the authors be sure that the measured XLD data is not affected by surface relaxation effects and fully*
reflecting bulk properties.

158

159 Thank you for this excellent point. We were initially also worried about problems with the surface. In general, we found that
 160 bulk properties (magnetic susceptibility, INS, x-ray diffraction, and transport) are all consistent with a temperature scale $T=60$
 161 K, which gives us confidence that the surface-sensitive measurements are seeing something very closely equivalent to what is
 162 found in bulk.

163 In Ref. 33 in the paper (Chikina, A. et al. Valence instability in the bulk and at the surface of the antiferromagnet SmRh₂Si₂
 164 Si₂. Phys. Rev. B 95, 155127 (2017)), the authors found that for the case of SmRh₂Si₂, valence fluctuations of the Sm atoms
 165 were the same in the bulk and at the surface. Since the crystal structure and general intermetallic environment of Sm is highly
 166 similar between SmRh₂Si₂ and SmCoIn₅, we expect that the valence fluctuations are stabilized by similar types of mechanisms
 167 in both materials. If the material is sensitive enough to the electronic structure to exhibit a valence instability, while exhibiting
 168 no surface-related effects, we expect that the properties of Sm are highly stable in such environments. We speculate that this
 169 might be due to the low anisotropy of the crystal field ground state wavefunction of Sm³⁺ in these materials.

170 Since the surface can in principle also be affected by surface modification, such as oxidation or contamination, we carefully
 171 prepared SmCoIn₅ samples in three separate ways for the XAS measurements. First, we prepared as-grown samples with no
 172 surface preparation; second, we cleaved samples before introducing the samples into the vacuum chamber; third, we cleaved the
 173 samples in a helium glove box before transferring them into the experimental chamber. We collected XAS scans on all three
 174 samples and searched for changes in the spectra under these conditions, or a reduction of the intensity, but we found that all three
 175 were equivalent. This shows that the sample surface is not strongly affected by any outside contamination, and is extremely
 176 stable and inert. In addition to making measurements generally easier to perform, this result also gives us confidence that the
 177 surface is less likely to be highly sensitive to additional electronic instabilities (for instance, a structural surface reconstruction,
 178 or surface Peierls instability).

179 Finally, the soft XAS work performed on the other materials CeCoIn₅, CeRhIn₅, CeIrIn₅, and their substitution series, a
 180 surface contribution was excluded on the basis of the same sample preparation steps that we performed. For example, in Ref. 39
 181 of the text (Willers, T. et al. Crystal-field and Kondo-scale investigations of Ce M In₅ (M = Co, Ir, and Rh): A combined x-ray
 182 absorption and inelastic neutron scattering study. Phys. Rev. B 81, 195114 [2010]), the authors state “We exclude depolarization
 183 effects due to surface degrading...since we recleaved and repeated the measurements to assure reproducibility”. We are also
 184 not aware of any claims about ARPES experiments on Ce-based 115s where surface effects were used to exclude any data.
 185 Therefore, we are quite confident that our soft x-ray experiments are probing close to bulk values.

186 To clarify this point, we have added the following lines in into the Methods section in the description of the XAS methods:

187 To ensure consistent sample preparation, we used three methods and tested all of them for differences: first, we prepared
 188 as-grown samples with no surface preparation; second, we cleaved samples before introducing the samples into the vacuum
 189 chamber; third, we cleaved the samples in a helium glove box before transferring them into the experimental chamber. We
 190 collected XAS scans on all three samples and searched for changes in the spectra under these conditions, or a reduction of the
 191 intensity, but we found that all three were equivalent; therefore, we rule out any surface-specific effects in our data.

192 *2. Can the authors show the XAS spectra, from which the XLD in Fig. 3 was derived, as a function of temperature. Do the*
 193 *authors see a growing Sm²⁺ below T_v and no Sm²⁺ above T_v in agreement with their model and consistent with the temperature*
dependence observed in the photoemission?

194

195 Thank you for this very important point. We conducted this analysis at an early stage, and were surprised that we do not see
 196 any direct signature of the Sm²⁺ component in the raw XAS spectra, which should appear most obviously as a shoulder near
 197 the left-hand side of the M₅ edge. We interpret this to mean that most likely the actual valence state is close to 3+ even in the
 198 low-temperature regime. Another explanation could be that the Sm²⁺ states are sufficiently delocalized that they do not exhibit

199 a strong signature in spectroscopy (the Sm²⁺ configuration must capture an electron from the metallic Fermi sea). In addition,
200 the lack of Sm²⁺ signature in the XLD can be explained because Sm²⁺ has J=0, which should be isotropic.

201 *3. If one accepts that the XLD data is purely bulk, how can the authors be sure that there is a unique solution for the CEF to
202 explain the observed XLD vs T?*

203 *3. Similar to the XAS data, can the authors comment on possible effects of large surface sensitivity in the PES data? Hyb-
204 dridisation phenomena, CEF effects and even 4f ionisation states at the surface can be different to that in the bulk. Different
205 surface terminations can completely change photoemission spectra even in the soft X-ray range. How can the authors be sure
that their spectra show bulk properties, and not a mixture of surface and bulk properties?*

206
207 We hope that we have sufficiently addressed the questions about the uniqueness of the CEF models in our reply to Referee 1
208 above.

209 Regarding the surface sensitivity of the ARPES data, we believe that the data reflect bulk values for the same reasons as with
210 the XAS data, since the range of photon energies was the same (Sm M_{4,5} edges).

211 *4. Can the excitation at 16 meV in the SmIrIn₅ INS also be due to a single CEF level with a second level either close to zero
212 or at higher energy. If the analogy to the CeMIn₅ (M=Co,Rh,Ir) holds, one would expect a smaller splitting between the two G7
213 levels than between the excited G7 and the G6. From a visual inspection the width of the 16 meV peak seems compatible with
214 one single excitation and the line shapes used to fit this peak seem narrower than those fitting the other peaks. Can the authors
comment? Do the presented INS data exclude a quasi quartet ground-state or another CEF 30 meV?*

215
216 Thank you for this point about the INS experiments. The energy resolution of the Eiger experiments was about 2.5 meV
217 near E=20 meV, making it unlikely that the INS features can arise only from a single peak. From the J=5/2 manifold in D_{4h}
218 symmetry, we expect two excited Kramers doublet levels. The only other levels would come from the J=7/2 manifold near
219 E=180 meV, which we do not expect to observe.

220 From the INS data, it is certainly possible that one crystal field level sits just above E=0, near E=2 meV. However, if this
221 were the case, we would expect a low-temperature feature in the magnetization data (T near 20 meV), which we do not see.
222 Furthermore, finding two doublets near E=0 would imply that these levels belonging initially to the Γ_8 quartet in Oh symmetry
223 which has been just slightly lifted by the D_{4h} symmetry. If this is the case, the shape of the XLD pattern would be completely
224 different (assuming SmCoIn₅ and SmIrIn₅ are roughly similar), and would be very unlikely to show an inversion of the pattern
225 near T=60 K.

226 From the analogy with CeCoIn₅, we believe that the meaningful comparison should be made with the Stevens operators, not
227 the level scheme. In other words, the Stevens operators reflect the actual electric potential of the environment surrounding the
228 rare earth ion, whereas the levels can be different between different rare earth ions due to their internal properties. In Fig. 2 of
229 the paper, we show that the Stevens operators take similar values except for the dipole A₂₀ parameter, which is likely strong in
230 Ce-based materials because of the long extension of the ground state wavefunction along the c-axis, which is the quantization
231 axis. With the addition of the on-site Coulomb repulsion U, as explained in the beginning of the supplementary information
232 and SFig. 1, the level scheme of SmCoIn₅ actually becomes reversed from what one would expect for the single-electron case.
233 Therefore, we believe that the INS results are consistent with the expectations of the crystal field model we present.

234 *5. I can follow the authors' argument, that the crystal field in SmCoIn₅ and SmIrIn₅ is likely not very different, based on
235 the very similar CEF schemes in the CeMIn₅ systems. As INS on SmCoIn₅ was not possible, the SmIrIn₅ experiment seems
236 meaningful also to discuss SmCoIn₅. However, in order to connect with and give confidence in the SmCoIn₅ XLD results, XLD
measurements on SmIrIn₅ would have been very important. Did the authors try to do such measurements?*

237
238 We agree that XAS experiments on SmIrIn₅ is a very interesting idea, and we also considered such experiments. Unfortunately,
239 the XAS data is time-consuming to collect, and we did not have time in our experiments to measure SmIrIn₅. Moreover, the
240 SmIrIn₅ sample preparation would also be much more challenging because the samples are more 3D-shaped with rounded faces,
241 and do not have a nicely cleavable surface. Therefore, we decided not to attempt these experiments ourselves.

242 *6. The susceptibility data seems very poorly fit but the proposed CEF model, not only below but also above 100K. Can the
243 authors show the inverse susceptibility and the fit with their CEF model in order to better judge the high-T region where the CEF
model should fit the data nicely.*

244
245 We hope that our response to Referee 1 above addresses the goodness of fit. As we wrote above, we believe that the "poorness"
246 of the fit is actually evidence supporting our key message about the existence of Sm delocalization.

247 We show in Fig. 3 the inverse of the susceptibility, which we agree is useful to tell that the experimental vales do not deviate
 248 much from the Curie-Weiss fit. The offset between the experimental data and crystal field model at high T can be due to the
 249 fact that the T-independent susceptibility was already subtracted from the experimental data, and may have some slight errors.
 250 The T-independent part of χ arises from the varnish, the natural diamagnetism of the elements in the sample, and the Van Vleck
 251 susceptibility which is large in Sm and Eu.

252 However, we believe that this figure does not provide any more insight compared to the raw data shown in the paper, and in
 253 addition could even be confusing because of the overlap of all 5 curves. For these reasons, we prefer to show the χ directly.

FIG. 3. Inverse of magnetic susceptibility curves shown in the paper.

254 7. In the low temperature region, the proposed CEF model deviates dramatically and qualitatively from the data. The
 255 susceptibility data reported here is very similar to that seen in CeMnIn5 and there a good correspondence between the measured
 256 susceptibility and its in-plane to out of plane anisotropy with the proposed CEF models was found. The authors invoke an
 257 emerging Co magnetic moment at low T due to 4f transfer into Co d states as the reason. I am not convinced that this is actually
 258 explain the observed deviation between their model and the data.

259 a) Could the temperature dependence of the XMCD detected Co moments also be explained as the response of constant,
 260 paramagnetic Co moments at fixed field vs T. Have the authors measured $M(H;T)$ to be sure that moments are saturated at 7T at
 261 least between 20 and 70K?

262 b) The XMCD data was measured at 7T, but the susceptibility at 0.5 T. It is hard to relate the two measurements without $M(H)$
 263 information. Similar to the XLD there's also the problem of surface vs bulk sensitivity when comparing these to techniques. Can
 264 the authors comment please.

265 c) What is the uncertainty on the extracted Co moments? The XMCD does not seem to vanish above the Co edge in SFig. 6.

266 d) Is the size of the detected Co moments (0.4 μ_B at 30K) compatible with the missing 0.4 to 0.5 emu/mol in the susceptibility?
 267 The estimated Co moments seem to large

268 e) How do the authors explain the directional dependence in Fig. 4c. Note that the field in the XMCD measurements was
 269 along 001 while the data points follow the 110 rather than the 001 susceptibility data.

270 f) The same poor fit to the low T region is also seen for SmIrIn5. Do the authors invoke the same explanation for the poor
 271 correspondence between their CEF model and the data in the Ir case, namely that 4f states start to delocalise into Ir 5d states and
 272 cause a finite 5d magnetic moment to just the right amount to recover very conventional Curie-Weiss behaviour in macroscopic
 measurements?

273

274 Overall, the magnetization measurements on CeCoIn5 do clearly show a “shoulder” near T=50 K, which is consistent with
 275 the Γ_7 ground state; see Fig. 4, which is reproduced from Shishido, H. et al. Fermi Surface, Magnetic and Superconducting
 276 Properties of LaRhIn5 and CeTlIn5 (T: Co, Rh and Ir). J. Phys. Soc. Jpn. 71, 162–173 (2002). We claim that SmCoIn5 also
 277 exhibits the same Γ_7 ground state, and should in principle also exhibit the shoulder, but that it is difficult to observe because of
 278 the delocalization of Sm states.

279 (a-b) We do believe that the Co moments may exhibit a small paramagnetic moment that saturates near $H=0.2$ T. We show
 280 M/H vs. H in Fig. 5, which is largely flat above $H=0.2$ (reflecting the paramagnetic Sm moments), but which deviates near
 281 $H=0$ which we believe arises from paramagnetic Co moments. We emphasize that the “Co Morb + Mspin” is a reflection of the
 282 intrinsic moment of Co combined with the delocalized Sm moments, which are temperature dependent.

283 (c) The XMCD raw data is shown supplementary SFig. 6. The XMCD requires integrating this data over both edges. Given
 284 a certain degree of noise in the raw data, we found it difficult to calculate an error bar in a meaningful way, especially at higher
 285 temperatures when the intensity is lower. Therefore, we plotted the data in a way to overlap with the susceptibility and show that
 286 the trend is the same.

287 (d) The measurements are not directly comparable, since the magnitude of the “Co Morb + Mspin” reflects the paramagnetic
 288 itinerant Sm moments. At $H=7$ T, this moment should be 14x larger than at $H=0.5$ T.

289 (e) We also conducted angle dependence of the XMCD signal, and found no change with angle, suggesting that there is no
 290 anisotropy of the delocalized itinerant moments.

291 (f) Our interpretation is that SmIrIn_5 is the same. However, since we only have INS and magnetization measurements on
 292 SmIrIn_5 , we think that our data serves a better purpose supporting the claims about SmCoIn_5 , rather than standing as independent
 293 claims about SmIrIn_5 . Due to the somewhat more challenging nature of the SmIrIn_5 samples, such as the lack of an easy cleavage
 294 plane, we chose not to continue our studies of SmIrIn_5 for this work.

FIG. 4. Magnetic susceptibility of CeCoIn_5 (Shishido 2002).

FIG. 5. $M(H)$ of SmCoIn_5 .

295 *7. I find the paragraph starting from line 438 several times, especially the part starting line 454, difficult to understand. Can
 the authors rephrase the main message of this section please.*

296 Thank you for this suggestion, we have rewritten the paragraph for clarity, as follows:

297 The energy difference observed between the Sm^{2+} component and the Fermi energy E_F observed in our Res-ARPES spectra
 298 raises the interesting possibility that some of the atomic multiplets of Sm play a role in the efficiency of the microscopic f
 299 electron delocalization mechanism in 115 materials. While the appearance of the Sm^{2+} level near $E_B = 0.4$ eV, marked by
 300

301 a dashed green line in Fig. 5(b), might be explained by a surface-core shift similar to SmSn_3 [31] and SmRh_2Si_2 [33], we
 302 also cannot discount the possibility that a multiplet level of Sm that is not the ground state participates in the Abrikosov-Suhl
 303 resonance process. In the Kondo effect, the width of the Kondo resonance is an indicator of the separation between the Fermi
 304 energy and the localized magnetic state. In CeCoIn_5 , CeRhIn_5 , and CeIrIn_5 , the width of the resonance indicates the Ce states
 305 to be less than 5 meV from E_F [39], while in SmB_6 , the Sm^{2+} states were shown to be 16 meV below E_F [54]. These results
 306 suggest that the width of the Kondo resonance in rare earth intermetallics is of order meV, far smaller than the separation of ~ 6
 307 eV in SmCoIn_5 would allow for hybridization to proceed through the ground state of Sm^{3+} . However, the Coulomb repulsion
 308 between mutual f electrons in Sm^{3+} distribute the $\binom{14}{5} = 2002$ f levels over more than 16 eV, as diagrammed in Fig. 2(d),
 309 and the distribution of $\binom{14}{6} = 3003$ Sm^{2+} multiplet states is similarly large. This fact suggests that a multiplet level of Sm
 310 could be found within 10-15 meV of E_F and therefore have a much larger Kondo hybridization matrix element compared to
 311 the actual crystal field ground state wavefunction found in this work. This opens the intriguing possibility of multiplet structure
 312 participating in the Kondo effect at E_F .

313 *8. Can the observed deviation from linear behaviour in the Hall effect also be explained within a CEF model without invoking
 hybridization (see e.g. J. Low Temp. Phys. 22, 665, 1976)?*

314

315 Thank you for this interesting suggestion. As we understand the reference you cited, the deviation of the Hall resistance in
 316 this case is due to an anomalous Hall coefficient with origins in magnetism. We initially had assumed that the Hall resistance
 317 in SmCoIn_5 would contain an anomalous component, however, after fitting the entire shape of the $R_H(H)$ curves, we found a
 318 much better agreement with an “ordinary” Hall effect within a two-band model, even at $T=2$ K in the magnetically ordered state.
 319 We show the agreement of both models in Fig. 6. Therefore, if the anomalous Hall effect is present in SmCoIn_5 , it will take an
 320 extraordinary effort to detect it. Based on the evidence from ARPES and other probes, we find it much more likely that a change
 321 in carrier density arises from a valence crossover.

Comparison of two different data models to analyze the Hall effect.

FIG. 6. Comparison of the two-band model with a model containing the anomalous Hall component.

How do the authors interpret the dispersing excitation in the INS data?

322

323 This is an interesting question, and we think that it might relate to a spurious feature in the measurement, such as an aluminum
 324 spurion.

REPLY TO REFEREE 3

325

326 *The authors explored the Sm ion's charge fluctuation in SmCoIn₅ and found evidence for the delocalization of Sm 4f electrons*
 327 *below 60 K. A lot of characterizations were conducted, making the analyses complete and results convincing. I recommend it*
 328 *publish in CommsPhys. Below are my comments/questions to the authors.*

328

329 We greatly appreciate the referee's feedback about the completeness of the analysis.

330 *1) In Figure 1, (a) and (b) should be marked on the left and right panels of the figure, respectively.*

331 *2) Line 133, "n_f" in the parentheses should be "Δn_f".*

332 *3) Lines 294-297: I guess the authors actually mean "...deviate from the experimental results"?*

332

333 We very much appreciate the careful reading, and we have fixed all three of these typos.

334 *4) The authors show Res-ARPES and XAS of SmCoIn₅ below 60 K. In order to support their claim that the Sm is mainly in its*
 335 *trivalent state above 60 K, it would be better to give the relevant data above 60 K. Do they just show a single peak for Sm³⁺?*

335

336 To describe the data above 60 K, we believe that the bulk probes (magnetization and, specifically, electrical resistivity) show
 337 that the valence state must be fairly constant. The XAS data seems to be only sensitive to the Sm³⁺ component, as we described
 338 in our reply to Referee 2 above. Unfortunately, we did not have time to carry out ARPES measurements at high temperature.
 339 However, given that we do not see a clear Sm²⁺ component in the raw XAS spectra, as we wrote in our reply to Referee 1 above,
 340 suggests that the valence is fairly close to 3 at all temperatures.

341 *5) It was claimed that there is an "upturn" in resistivity vs temperature curve below 60 K. However, it does not look like an*
 342 *upturn. It is just gradually deviating from R/T, which is quite common in even nonmagnetic metals because at lower temperatures*
 343 *resistivity is dominated by the T⁵ term of the phonon scattering. Not sure if it makes sense to divide R(T) into two regions by 60*
 344 *K.*

344

345 Thank you for drawing attention to this phenomenon. In general, we agree that the electrical resistivity should also contain
 346 scattering terms that reflect phonons, but we don't know any examples reported in 115 materials and closely related materials.
 347 For example, in CeCoIn₅ we are not aware that phonon scattering has been reported, and a careful analysis of the resistivity
 348 in a recent reference showed that there is no contribution from phonons expected until temperatures above T=100 K (Jang, S.
 349 et al. Evolution of the Kondo lattice electronic structure above the transport coherence temperature. PNAS 117, 23467–23476
 350 (2020)). In our data, we find a completely T-linear trend up to T=300 K, which suggests that phonon scattering effects must be
 351 quite small.

352 Moreover, in Ref. 25 in the paper (Kasaya, M. et al. Quadrupolar ordering and dense kondo behaviour in SmSn₃. Journal
 353 of Magnetism and Magnetic Materials 52, 289–292 (1985)), electrical resistivity data is presented for SmSn₃ and SmIn₃. In
 354 SmSn₃, the authors found an "upturn" exists for SmSn₃, but not for SmIn₃, which they argue is distinct evidence for a Kondo
 355 effect in SmSn₃. The absence of a similar phenomenon in SmIn₃ shows that this effect cannot arise from phonon scattering.

356 Therefore, while we agree that "upturn" is not necessarily the obvious term for describing the SmCoIn₅ data on its own, we
 357 believe that this term more generally classifies the departure from T-linear trend as belonging to a family of compounds which
 358 display incoherent Kondo scattering.

FIG. 7. Electrical resistivity of CeCoIn_5 (Jang 2020).

FIG. 8. Electrical resistivity of SmSn_3 and SmIn_3 (Kasaya 1985).

REVIEWERS' COMMENTS:

Reviewer #1 (Remarks to the Author):

I have studied the "rebuttal" document in detail. The authors have given due attention to the questions that I had raised and have provided thorough responses.

The level of the explanations demonstrates that the experimental work was carried out with care and the data was analysed with diligence.

Where applicable, the text and figures have been adequately improved.

As indicated in my original review, I recommend publication.

Reviewer #2 (Remarks to the Author):

I can recommend the revised version of the manuscript for publication in Comm. Phys.